# Clathrin-independent endocytic retrieval of SV proteins mediated by the clathrin adaptor AP-2 at mammalian central synapses

**Tania López-Hernández[1†], Koh-ichiro Takenaka[2†], Yasunori Mori[2], Pornparn Kongpracha[3], Shushi Nagamori[3], Volker Haucke[1]\*, Shigeo Takamori[2]\***

[1]Leibniz-Forschungsinstitut für Molekulare Pharmakologie (FMP), Berlin, Germany; [2]Laboratory of Neural Membrane Biology, Graduate School of Brain Science, Doshisha University, Kyoto, Japan; [3]Department of Laboratory Medicine, The Jikei University School of Medicine, Tokyo, Japan

**\*For correspondence:**
haucke@fmp-berlin.de (VH);
stakamor@mail.doshisha.ac.jp
(ST)

[†]These authors contributed
equally to this work

**Competing interest:** The authors declare that no competing interests exist.

**Abstract** Neurotransmission is based on the exocytic fusion of synaptic vesicles (SVs) followed by endocytic membrane retrieval and the reformation of SVs. Conflicting models have been proposed regarding the mechanisms of SV endocytosis, most notably clathrin/adaptor protein complex 2 (AP-2)-mediated endocytosis and clathrin-independent ultrafast endocytosis. Partitioning between these pathways has been suggested to be controlled by temperature and stimulus paradigm. We report on the comprehensive survey of six major SV proteins to show that SV endocytosis in mouse hippocampal neurons at physiological temperature occurs independent of clathrin while the endocytic retrieval of a subset of SV proteins including the vesicular transporters for glutamate and GABA depend on sorting by the clathrin adaptor AP-2. Our findings highlight a clathrin-independent role of the clathrin adaptor AP-2 in the endocytic retrieval of select SV cargos from the presynaptic cell surface and suggest a revised model for the endocytosis of SV membranes at mammalian central synapses.

## Editor's evaluation

Neuronal function requires the recycling of synaptic vesicle proteins from the presynaptic plasma membrane to reform synaptic vesicles. This study highlights a clathrin-independent role of the clathrin adaptor, AP-2 in the endocytic retrieval of select synaptic vesicle cargos at mammalian central synapses. The work will be of interest to cell biologists and neurobiologists interested in understanding the molecular basis of neurotransmission.

## Introduction

Synaptic transmission relies on the release of neurotransmitters by calcium-triggered exocytic fusion of synaptic vesicles (SVs), tiny organelles (~40 nm in diameter) that store and secrete neurotransmitter molecules at specialized active zone (AZ) release sites within presynaptic nerve terminals (*Südhof, 2004*). Following exocytosis, SVs are locally reformed via compensatory endocytic retrieval of membrane and its protein constituents (i.e., SV proteins) from the cell surface in order to keep the presynaptic surface area constant and to ensure sustained neurotransmission (*Dittman and Ryan, 2009*; *Kononenko and Haucke, 2015*). While the core components that mediate the exo- and endocytosis of SVs have been identified and characterized in detail (*Rizzoli, 2014*; *Takamori et al., 2006*;

*Wilhelm et al., 2014*), the molecular mechanisms underlying SV endocytosis and reformation have remained controversial (*Delvendahl et al., 2016*; *Kononenko et al., 2014*; *Soykan et al., 2017*; *Watanabe et al., 2013*; *Watanabe et al., 2014*; *Milosevic, 2018*).

Pioneering ultrastructural analyses of stimulated frog neuromuscular junctions using electron microscopy (EM) suggested that SVs are recycled via clathrin-mediated endocytosis (CME) of plasma membrane infoldings or budding from cisternal structures located away from the AZ (*Heuser and Reese, 1973*). Subsequent studies in neurons and in nonneuronal models showed that CME occurs on a timescale of many seconds and crucially depends on clathrin and its essential adaptor protein complex 2 (AP-2) (*Mitsunari et al., 2005*), a heterotetramer comprising α, β2, μ2, and σ2 subunits, as well as a plethora of endocytic accessory proteins (*Südhof, 2004*; *Dittman and Ryan, 2009*; *Kononenko and Haucke, 2015*; *Rizzoli, 2014*; *Saheki and De Camilli, 2012*; *Kaksonen and Roux, 2018*). These and other works led to the view that SVs are primarily, if not exclusively, recycled by clathrin/AP-2-dependent CME (*Granseth et al., 2006*; *Granseth et al., 2007*).

Recent studies using high-pressure freezing EM paired with optogenetic stimulation have unraveled a clathrin-independent mechanism of SV endocytosis (CIE) in response to single action potential (AP) stimuli that selectively operates at physiological temperature (*Watanabe et al., 2013*). This ultrafast endocytosis (UFE) pathway is distinct from kiss-and-run or kiss-and-stay exoendocytosis observed in neuroendocrine cells (*Alés et al., 1999*; *Shin et al., 2018*), operates on a timescale of hundreds of milliseconds, and results in the generation of endosome-like vacuoles (ELVs) from which SVs can reform via clathrin-mediated budding processes (*Kononenko et al., 2014*; *Watanabe et al., 2014*). Temperature-sensitive, clathrin-independent SV endocytosis has also been observed by presynaptic capacitance recordings at cerebellar mossy fiber boutons and by optical imaging at small hippocampal synapses (*Delvendahl et al., 2016*; *Kononenko et al., 2014*; *Soykan et al., 2017*) and is compatible with the accumulation of postendocytic presynaptic vacuoles upon acute or sustained genetic perturbation of clathrin at stimulated fly neuromuscular junctions (*Kasprowicz et al., 2014*; *Kasprowicz et al., 2008*; *Heerssen et al., 2008*) and at mammalian central synapses (*Kononenko et al., 2014*; *Imig et al., 2020*). Collectively, these studies suggest that SV endocytosis under physiological conditions is primarily mediated by CIE (e.g., UFE), while the function of clathrin and clathrin adaptors such as AP-2 is limited to the reformation of functional SVs from internal ELVs rather than acting at the plasma membrane proper.

While this model can provide a mechanistic explanation for the observed speed of SV endocytosis, the key question of how SV proteins are sorted to preserve the compositional integrity of SVs (*Takamori et al., 2006*) remains unresolved. Four points pertaining to this question are to be considered. First, optical imaging-based acid quench experiments in hippocampal neurons indicate that the capacity of UFE is limited to single or few APs, while the majority of SV proteins appear to be internalized on a timescale of several seconds following AP train stimulation (*Dittman and Ryan, 2009*; *Soykan et al., 2017*; *Kim and Ryan, 2009*), that is a timescale compatible with either CME or CIE. Second, as high-pressure freezing EM experiments have not been able to reveal the fate and time course of endocytosis of SV proteins, it is formally possible that UFE proceeds under conditions of clathrin depletion (*Watanabe et al., 2014*), while SV proteins remain stranded on the neuronal surface. Third, mutational inactivation of the binding motifs for the clathrin adaptor AP-2 in the vesicular transporters for glutamate (VGLUT) and γ-aminobutyric acid (VGAT) severely compromises the speed and efficacy of their endocytic retrieval at room temperature (*Voglmaier et al., 2006*; *Santos et al., 2013*; *Li et al., 2017*; *Foss et al., 2013*), arguing that at least under these conditions (e.g., low temperature when UFE is blocked) these SV proteins may be retrieved from the cell surface via clathrin/AP-2. Finally, genetic inactivation of clathrin/AP-2-associated endocytic adaptors for the sorting of specific SV proteins, for example Stonin 2, an adaptor for the SV calcium sensor Synaptotagmin, or AP180, an adaptor for Synaptobrevin/VAMP2, causes the accumulation of their respective SV cargos at the neuronal plasma membrane (*Kononenko and Haucke, 2015*; *Cousin, 2017*; *Kaempf et al., 2015*; *Koo et al., 2015*; *Mori and Takamori, 2017*). These data could be interpreted to indicate that at least some SV proteins are endocytosed via CME, whereas others may use CIE mechanisms such as UFE. However, such a model bears the problem of how CME and CIE pathways are coordinated and how membrane homeostasis is then maintained.

To solve the question how SV protein sorting is accomplished and how this relates to CME- *vs* CIE-based mechanisms for SV endocytosis, we have conducted a comprehensive survey of six major SV

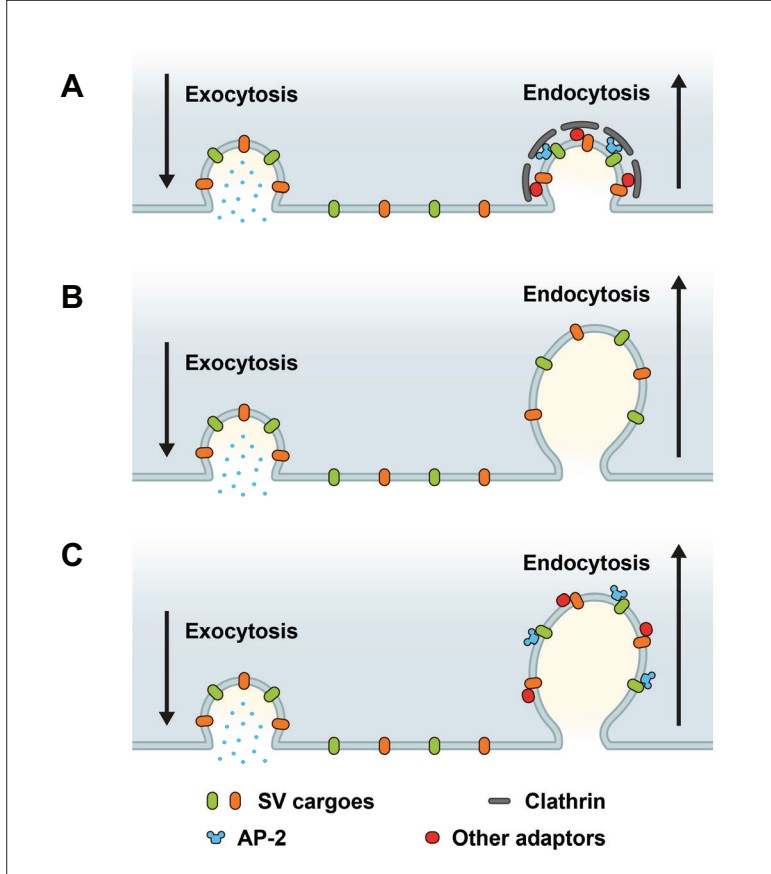

**Figure 1.** Possible roles of clathrin and adaptor protein complex 2 (AP-2) in synaptic vesicle (SV) endocytosis and SV cargo retrieval. (**A**) A model predicting that SV retrieval following neurotransmitter release is mediated by clathrin-mediated endocytosis (CME) where AP-2 functions as bridge between SV cargos and clathrin to form clathrin-coated pits (CCPs) at plasma membrane. In this scenario, inactivation of both clathrin and AP-2 would slow either SV endocytosis as well as SV cargo retrieval. (**B**) A model predicting that SV endocytosis occurs in a clathrin-independent manner (CIE), and neither clathrin nor AP-2 mediate SV endocytosis and SV cargo retrieval at plasma membrane. If this were the case, inactivation of both clathrin and AP-2 would not change the kinetics rate of SV endocytosis and SV cargo retrieval. (**C**) A model predicting that dedicated adaptors such as AP-2 function as sorting protein for SV cargo even during CIE. If this were the case, inactivation of clathrin and AP-2 would produce distinct phenotypes between SV endocytosis and SV cargo retrieval.

proteins in primary hippocampal neurons depleted of clathrin or conditionally lacking AP-2. We show that clathrin is dispensable for the endocytosis of all SV proteins at physiological temperature independent of the stimulation paradigm. In contrast, endocytic retrieval of a subset of SV proteins including VGLUT1 and VGAT depends on sorting by AP-2. Our findings highlight a clathrin-independent function of the clathrin adaptor AP-2 in the endocytic retrieval of select SV cargos from the presynaptic plasma membrane and suggest a revised model for SV endocytosis and recycling.

## Results

Based on prior works (*Dittman and Ryan, 2009*; *Rizzoli, 2014*; *Delvendahl et al., 2016*; *Kononenko et al., 2014*; *Soykan et al., 2017*; *Watanabe et al., 2014*; *Milosevic, 2018*; *Heuser and Reese, 1973*; *Saheki and De Camilli, 2012*; *Takei et al., 1996*), three main models for the sorting and endocytic recycling of SV proteins at central mammalian synapses can be envisaged (*Figure 1*). According to the classical CME-based model of SV endocytosis, SV proteins exocytosed in response to AP trains undergo clathrin/AP-2-mediated sorting and endocytosis from the presynaptic plasma membrane or plasma membrane infoldings (*Takei et al., 1996*) akin to CME in receptor-mediated

endocytosis in nonneuronal cells (*Kaksonen and Roux, 2018*). This model predicts that loss of either clathrin or its essential adaptor AP-2 delays the endocytic retrieval of all major SV proteins (*Figure 1A*). A second model supported by elegant high-pressure freezing (*Watanabe et al., 2014*), electrophysiological (*Delvendahl et al., 2016*), and optical imaging (*Kononenko et al., 2014*; *Soykan et al., 2017*) experiments suggests that exocytosed SV proteins are internalized via clathrin- and AP-2-independent bulk endocytosis. In this model, SV protein sorting occurs from internal ELVs that are formed downstream of the endocytic internalization step. Hence, at physiological temperature the endocytic retrieval of all major SV proteins would proceed unperturbed in the absence of either clathrin or AP-2 (*Figure 1B*). Finally, it is conceivable that exocytosed SV proteins present on the neuronal surface are sorted by dedicated endocytic adaptors, for example the AP-2 complex, to facilitate their clathrin-independent internalization via CIE. Clathrin, possibly in conjunction with AP-2 and other adaptors then operates downstream of CIE to reform functional SVs from ELVs. In this case, loss of clathrin or AP-2 is predicted to result in distinct phenotypes: While endocytosis of SV proteins is unperturbed upon depletion of clathrin, loss of AP-2 would be expected to selectively affect the rate and efficacy of endocytosis of distinct SV cargos recognized by AP-2 (*Figure 1C*).

## Endocytic retrieval of SV proteins in hippocampal neurons occurs independent of clathrin at physiological temperature

To distinguish between these models, we optically recorded the stimulation-induced exo-/endocytosis of SV proteins carrying within their luminal domains a pH-sensitive superecliptic green fluorescent protein (SEP, often also referred to as pHluorin) that is dequenched during exocytosis and undergoes requenching as SVs are internalized and reacidified (*Miesenböck et al., 1998*; *Sankaranarayanan et al., 2000*). Specifically, we monitored SEP-tagged chimeras of the calcium sensor Synaptotagmin 1 (Syt1), the multispanning glycoprotein SV2A, the SNARE protein Synaptobrevin/VAMP2 (hereafter referred to as Syb2), the tetraspanin Synaptophysin (Syp), the vesicular glutamate transporter 1 (VGLUT1), and the vesicular GABA transporter (VGAT), which have been used extensively to monitor SV recycling in various preparations (*Kononenko et al., 2014*; *Soykan et al., 2017*; *Granseth et al., 2006*; *Kim and Ryan, 2009*; *Voglmaier et al., 2006*; *Miesenböck et al., 1998*; *Sankaranarayanan et al., 2000*) and constitute the major protein complement of SVs based on their copy numbers (*Takamori et al., 2006*). We capitalized on the fact that, in hippocampal neurons stimulated with trains of APs, SV endocytosis occurs on a timescale of >10 s at physiological temperature (*Soykan et al., 2017*), for example a timescale that is much slower than requenching of SEP due to reacidification of newly endocytosed vesicles (*Atluri and Ryan, 2006*; *Egashira et al., 2015*). Therefore, under these conditions, the decay of SEP signals can serve as a measure of the time course of SV endocytosis.

We first depleted clathrin heavy chain (CHC) in hippocampal neurons using lentiviral vectors to ~10–25% of the levels found in controls as evidenced by confocal imaging of immunostained samples and by immunoblot analysis (*Figure 2—figure supplement 1A–D*), in agreement with previous data (*Kononenko et al., 2014*; *Watanabe et al., 2014*; *Granseth et al., 2006*). Lentiviral shRNA-mediated depletion of clathrin potently inhibited uptake of transferrin into cultured neurons, indicating effective blockade of CME (*Figure 2A, B*). To assess the effects of clathrin loss on the stimulation-induced endocytic retrieval of SV proteins, we stimulated control or clathrin-depleted hippocampal neurons expressing any one of the six major SEP-tagged SV proteins with a high-frequency stimulus train (200 APs applied at 40 Hz) at physiological temperature (35 ± 2°C), and monitored fluorescence rise and decay over time. Strikingly, exo-/endocytosis of all SEP-tagged SV proteins proceeded with unaltered kinetics, that is $\tau$ ~ 15–20 s, irrespective of the depletion of clathrin (*Figure 2C–N*). Similar results were seen if clathrin function was acutely blocked by application of the small molecule inhibitor Pitstop2 (*von Kleist et al., 2011*; *Figure 2O, P*), a condition that potently inhibited CME of transferrin (*Figure 2—figure supplement 1E,F*). When these experiments were repeated under conditions of low-frequency stimulation (200 APs applied at 5 Hz) at room temperature (RT), that is conditions in which the efficacy of CIE is reduced (*Watanabe et al., 2014*), the endocytic retrieval of Syp-SEP (also often referred to as SypHy) or VGLUT1-SEP was delayed in neurons depleted of clathrin (*Figure 2—figure supplement 1G-J*), consistent with earlier data using Syt1-SEP as a reporter (*Kononenko et al., 2014*).

These results show that in small hippocampal synapses at physiological temperature, endocytosis of all major SV proteins and hence, of SVs as a whole, occurs independent of clathrin via CIE.

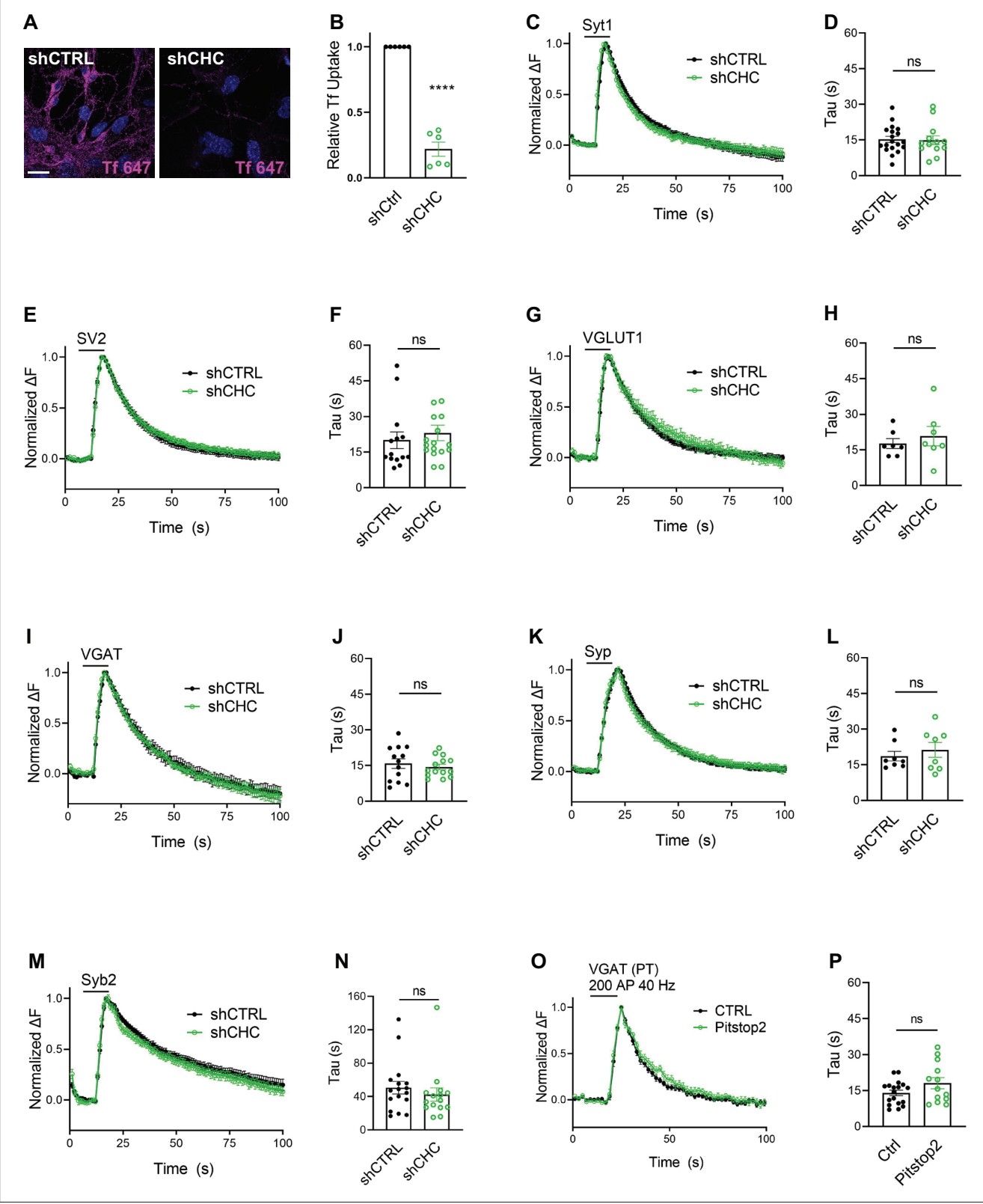

**Figure 2.** Synaptic vesicle (SV) endocytosis in hippocampal neurons occurs independent of clathrin at physiological temperature. (**A**) Representative images of primary neurons transduced with shCTRL or shCHC and allowed to internalize AlexaFluor[647]-labeled transferrin (Tf647) for 20 min at 37°C. The scale bar represents 20 μm. (**B**) Quantification of data shown in (**A**). Values for shCTRL were set to 1. The data represent mean ± standard error of the mean (SEM) from $n = 6$ independent experiments. ****$p = 0.0001$, two-sided one-sample $t$-test. (**C–N**) Average normalized SEP fluorescence traces

*Figure 2 continued on next page*

*Figure 2 continued*

of neurons transduced with lentivirus expressing nonspecific shRNA (shCTRL) or shRNA-targeting CHC (shCHC) and cotransfected with SEP probes tagged to the luminal portion of Syt1 (**C**), SV2A (**E**), VGLUT1 (**G**), VGAT (**I**), Syp (**K**), and Syb2 (**M**) subjected to electrical stimulation of 40 Hz (200 APs) at physiological temperature. Endocytic decay time constant ($\tau$) of transfected and lentivirally transduced neurons coexpressing, respectively, (**D**) Syt1-SEP and shCTRL (15.22 ± 1.30 s) or shCHC (14.87 ± 1.87 s); (**F**) SV2A-SEP and shCTRL (20.00 ± 3.53 s) or shCHC (23.06 ± 3.31 s); (**H**) VGLUT1-SEP and shCTRL (17.67 ± 2.05 s) or shCHC (20.79 ± 4.07 s); (**J**) VGAT-SEP and shCTRL (15.82 ± 2.04 s) or shCHC (14.38 ± 1.14 s); (**L**) Syp-SEP and shCTRL (18.65 ± 2.03 s) or shCHC (21.27 ± 3.10 s); and (**N**) Syb2-SEP and shCTRL (50.63 ± 7.49 s) or shCHC (42.32 ± 8.03 s). Data shown represent the mean ± SEM for Syt1 ($n_{CTRL}$ = 19 images, $n_{shCHC}$ = 13 images; p = 0.875), for SV2A ($n_{shCTRL}$ = 14 images, $n_{shCHC}$ = 17 images; p = 0.533), for VGLUT1 ($n_{shCTRL}$ = 7 images, $n_{shCHC}$ = 7 images; p = 0.506), for VGAT ($n_{shCTRL}$ = 13 images, $n_{shCHC}$ = 14 images; p = 0.534), for Syp ($n_{shCTRL}$ = 8 images, $n_{shCHC}$ = 8 images; p = 0.490), and for Syb2 ($n_{shCTRL}$ = 17 images, $n_{shCHC}$ = 15 images; p = 0.455). Two-sided unpaired *t*-test. (**O, P**) Endocytosis of VGAT upon acute inactivation of clathrin by Pitstop2 proceeds unaffected at physiological temperature. (**O**) Average normalized traces of neurons transfected with VGAT-SEP and treated either with DMSO (CTRL) or Pitstop2 in response to 200 APs applied at 40 Hz. (**P**) Endocytic decay time constant ($\tau$) of neurons expressing VGAT-SEP ($\tau_{CTRL}$ = 14.03 ± 1.16 s, $\tau_{Pitstop2}$ = 18.11 ± 2.32 s). Data shown represent the mean ± SEM with n = 18 images and n = 13 images for CTRL and Pitstop2, respectively. p = 0.0976. Two-sided unpaired *t*-test. Raw data can be found in *Figure 2—source data 1*.

The online version of this article includes the following source data and figure supplement(s) for figure 2:

**Source data 1.** Source data for *Figure 2B-P*.

**Figure supplement 1.** Temperature-sensitive, clathrin-independent synaptic vesicle (SV) endocytosis at hippocampal synapses.

**Figure supplement 1—source data 1.** Source data for *Figure 2—figure supplement 1B, D, F, G, H, I, J*.

**Figure supplement 1—source data 2.** Raw uncropped immunoblot image for *Figure 2—figure supplement 1C*.

## CIE of a subset of SV proteins depends on the clathrin adaptor AP-2

We next set out to analyze whether endocytosis of the major SV proteins is also independent of the essential clathrin adaptor complex AP-2. This would be expected, if SV endocytosis was mediated by CIE and the sole function of clathrin/AP-2 was to reform SVs from postendocytic ELVs (*Figure 1B*). We conditionally ablated AP-2 expression by tamoxifen induction of Cre recombinase in hippocampal neurons from *Ap2m1^lox/lox^* mice crossed with inducible CAG-Cre transgenic mice resulting in a reduction of AP-2 levels to <15% of that detected in WT control neurons (hereafter referred to as AP-2μ KO) (*Figure 3—figure supplement 1A,B*; *Kononenko et al., 2014*; *Soykan et al., 2017*). Further depletion below this level caused neuronal death.

Endocytosis of Syt1-SEP and SV2A-SEP proceeded with similar kinetics in control or AP-2μ KO hippocampal neurons stimulated with 200 APs applied at 40 Hz at physiological temperature, consistent with our earlier findings (*Kononenko et al., 2014*; *Soykan et al., 2017*; *Figure 3A–D*). Surprisingly, however, we found that loss of AP-2 significantly slowed down the endocytic retrieval of other major SV proteins such as Syp, Syb2, and most prominently, of the vesicular neurotransmitter transporters VGLUT1 and VGAT (*Figure 3E–L*). These phenotypes were specific as plasmid-based reexpression of AP-2μ in AP-2μ KO neurons rescued defective endocytosis of these SEP-tagged SV proteins (*Figure 3*).

As elevated pHluorin signals could conceivably arise from defects in endocytosis or vesicle reacidification, we used a quench protocol in which acidic buffer is applied before and after neuronal stimulation with 200 APs to probe the accessibility of VGLUT1- or SV2A-SEP to externally applied acid (*Figure 3M–P*). These experiments showed that exocytosed VGLUT1-SEP accumulates on the surface of AP-2μ KO neurons (*Figure 3M*) as quantitatively evidenced by an increased $\Delta F_2/\Delta F_1$ ratio (*Figure 3N*). No difference in the fraction of surface-accumulated SV2A-SEP molecules was observed in AP-2μ KO compared to WT neurons (*Figure 3O, P*). Defective endocytosis of VGLUT1-SEP in the absence of AP-2 was further confirmed by probing the acid-resistant pool of endocytosed VGLUT1-SEP at 30 s poststimulation with 200 APs in the presence of folimycin, a selective inhibitor of vesicle reacidification by the V-ATPase (*Figure 3—figure supplement 1C-F*). These experiments demonstrate that AP-2 is required for VGLUT1-SEP endocytosis but is dispensable for vesicle reacidification.

We challenged these data acquired with strong 200 AP stimulation by monitoring SV endocytosis in response to a milder stimulation paradigm that results in the exocytic fusion of the readily releasable pool of SVs (*Hua et al., 2011*; *Murthy and Stevens, 1999*), that is 50 APs applied at 20 Hz. While endocytosis of SV2A- and VGLUT1-SEP proceeded unperturbed in hippocampal neurons depleted of clathrin (*Figure 4A–D*), a substantial delay in the endocytosis of VGLUT1- but not SV2-SEP was observed in neurons lacking AP-2 (*Figure 4E–H*). No difference was found in the fraction of boutons responding to stimulation with 50 APs between WT and AP-2μ KO neurons (*Figure 4—figure*

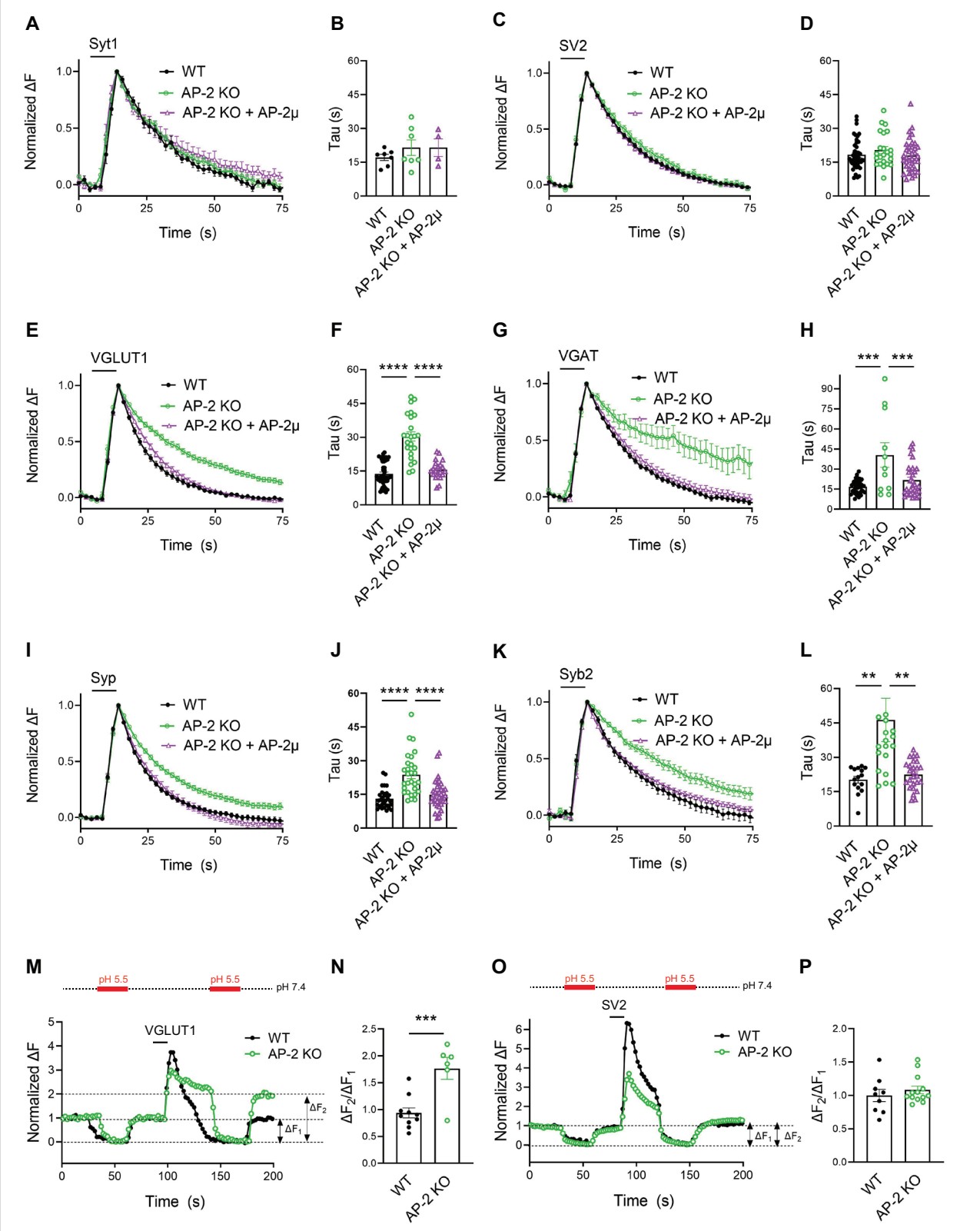

**Figure 3.** Clathrin-independent endocytic retrieval of select synaptic vesicle (SV) cargos by the clathrin adaptor adaptor protein complex 2 (AP-2) at physiological temperature. (**A–D**) Poststimulus retrieval of Syt1 and SV2A in the absence of AP-2 persists unaffected in response to 200 APs applied at 40 Hz. Average normalized traces of WT and AP-2μ KO derived neurons cotransfected with Syt1-SEP (**A**) or SV2A-SEP (**C**) and mRFP or rescued by reexpression of untagged AP-2μ subunit together with soluble mRFP (AP-2μ) to identify transfected neurons in response to 200 APs applied at 40 Hz.

*Figure 3 continued on next page*

*Figure 3 continued*

Quantification of the endocytic decay time constant ($\tau$) of neurons expressing Syt1-SEP (**B**) ($\tau_{WT}$ = 17.17 ± 1.35 s, $\tau_{AP\text{-}2\mu\ KO}$ = 21.53 ± 3.43 s, $\tau_{AP\text{-}2\mu\ KO+AP\text{-}2\mu}$ = 21.51 ± 3.91 s) or SV2A-SEP (**D**) ($\tau_{WT}$ = 18.33 ± 0.99 s, $\tau_{AP\text{-}2\mu\ KO}$ = 20.45 ± 1.56 s, $\tau_{AP\text{-}2\mu\ KO+AP\text{-}2\mu}$ = 18.04 ± 1.16 s). Data shown represent the mean ± standard error of the mean (SEM): Syt1 ($n_{WT}$ = 7 images, $n_{AP\text{-}2\mu\ KO}$ = 7 images, $n_{AP\text{-}2\mu\ KO+AP\text{-}2\mu}$ = 4 images; $p_{(WT\ vs\ AP\text{-}2\mu\ KO)}$ = 0.4994; $p_{(AP\text{-}2\mu\ KO\ vs\ AP\text{-}2\mu\ KO+AP2\mu)}$ > 0.9999); SV2A ($n_{WT}$ = 44 images, $n_{AP\text{-}2\mu\ KO}$ = 23 images, $n_{AP\text{-}2\mu\ KO+AP\text{-}2\mu}$ = 37 images; $p_{(WT\ vs\ AP\text{-}2\mu\ KO)}$ = 0.4665; $p_{(AP\text{-}2\mu\ KO\ vs\ AP\text{-}2\mu\ KO+AP\text{-}2\mu)}$ = 0.3943). One-way analysis of variance (ANOVA) with Tukey's post-test. (**E–L**) Loss of AP-2 significantly delay the endocytic retrieval of other major SV proteins. Average normalized traces of WT and AP-2μ KO neurons cotransfected with VGLUT1-SEP (**E**), VGAT-SEP (**G**), Syp-SEP (**I**), Syb2-SEP (**K**), and mRFP or AP-2μ to rescue AP-2μ expression stimulated with 200 APs at 40 Hz. Endocytic decay time constants ($\tau$) were calculated from WT, AP-2μ KO neurons, and AP-2μ KO neurons rescued by reexpression of AP-2μ expressing VGLUT1-SEP (**F**) ($\tau_{WT}$ = 13.52 ± 0.90 s, $\tau_{AP\text{-}2\mu\ KO}$ = 30.33 ± 2.01 s, $\tau_{AP\text{-}2\mu\ KO+AP\text{-}2\mu}$ = 16.38 ± 1.00 s), VGAT-SEP (**H**) ($\tau_{WT}$ = 16.76 ± 0.87 s, $\tau_{AP\text{-}2\mu\ KO}$ = 40.56 ± 9.21 s, $\tau_{AP\text{-}2\mu\ KO+AP\text{-}2\mu}$ = 21.77 ± 2.15 s), Syp-SEP (**J**) ($\tau_{WT}$ = 13.05 ± 0.74 s, $\tau_{AP\text{-}2\mu\ KO}$ = 23.76 ± 1.72 s, $\tau_{AP\text{-}2\mu\ KO+AP\text{-}2\mu}$ = 14.89 ± 1.04 s), VGAT-SEP (**D**) ($\tau_{WT}$ = 18.33 ± 0.99 s, $\tau_{AP\text{-}2\mu\ KO}$ = 20.45 ± 1.56 s, $\tau_{AP\text{-}2\mu\ KO+AP\text{-}2\mu}$ = 18.04 ± 1.16 s), and Syb2-SEP (**L**) ($\tau_{WT}$ = 20.27 ± 1.48 s, $\tau_{AP\text{-}2\mu\ KO}$ = 46.34 ± 9.43 s, $\tau_{AP\text{-}2\mu\ KO+AP\text{-}2\mu}$ = 22.46 ± 1.21 s). Data shown represent the mean ± SEM: VGLUT1 ($n_{WT}$ = 37 images, $n_{AP\text{-}2\mu\ KO}$ = 24 images, $n_{AP\text{-}2\mu\ KO+AP\text{-}2\mu}$ = 23 images; ****$p_{(WT\ vs\ AP\text{-}2\mu\ KO)}$ < 0.0001; ****$p_{(AP\text{-}2\mu\ KO\ vs\ AP\text{-}2\mu\ KO+AP\text{-}2\mu)}$ < 0.0001); VGAT ($n_{WT}$ = 34 images, $n_{AP\text{-}2\mu\ KO}$ = 11 images, $n_{AP\text{-}2\mu\ KO+AP\text{-}2\mu}$ = 32 images; ****$p_{(WT\ vs\ AP\text{-}2\mu\ KO)}$ < 0.0001; ***$p_{(AP\text{-}2\mu\ KO\ vs\ AP\text{-}2\mu\ KO+AP\text{-}2\mu)}$ = 0.0008); Syp ($n_{WT}$ = 33 images, $n_{AP\text{-}2\mu\ KO}$ = 29 images, $n_{AP\text{-}2\mu\ KO+AP\text{-}2\mu}$ = 37 images; ****$p_{(WT\ vs\ AP\text{-}2\mu\ KO)}$ < 0.0001; ****$p_{(AP\text{-}2\mu\ KO\ vs\ AP\text{-}2\mu\ KO+AP\text{-}2\mu)}$ < 0.0001); Syb2 ($n_{WT}$ = 15 images, $n_{AP\text{-}2\mu\ KO}$ = 20 images, $n_{AP\text{-}2\mu\ KO+AP\text{-}2\mu}$ = 26 images; **$p_{(WT\ vs\ AP\text{-}2\mu\ KO)}$ = 0.0083; **$p_{(AP\text{-}2\mu\ KO\ vs\ AP\text{-}2\mu\ KO+AP\text{-}2\mu)}$ = 0.0052). One-way ANOVA with Tukey's post-test. (**M–P**) Delayed poststimulus retrieval of major SV proteins in the absence of AP-2 is not caused by defects in re-acidification of endocytosed vesicles. Representative normalized traces of WT and AP-2μ KO neurons expressing VGLUT1-SEP (**M**) or SV2A-SEP (**O**) stimulated with 200 APs applied at 40 Hz and subjected to low pH imaging buffer before and after train stimulation. Fluorescence quenching by application of acidic buffer poststimulus ($\Delta F_2$) vs prestimulus ($\Delta F_1$) of VGLUT1-SEP (**N**) ($n_{WT}$ = 10 images, $n_{AP\text{-}2\mu\ KO}$ = 6 images) or SV2A-SEP (**P**) ($n_{WT}$ = 8 images, $n_{AP\text{-}2\mu\ KO}$ = 13 images) is taken as a measure to probe the SEP surface pool in WT and AP-2μ KO hippocampal neurons. Values for WT were set to 1. Data shown represent the mean ± SEM. VGLUT1: WT = 1.0 ± 0.1; AP-2μ KO = 1.8 ± 0.2. p = 0.0009. SV2: WT = 1.0 ± 0.1; AP-2μ KO = 1.1 ± 0.1. p = 0.4267. Two-sided unpaired *t*-test. Raw data can be found in *Figure 3—source data 1*.

The online version of this article includes the following source data and figure supplement(s) for figure 3:

**Source data 1.** Source data for *Figure 3A-P*.

**Figure supplement 1.** Adaptor protein complex 2 (AP-2) depletion in hippocampal neurons impairs endocytic retrieval of VGLUT1-SEP.

**Figure supplement 1—source data 1.** Source data for *Figure 3—figure supplement 1B, D, E, F*.

---

supplement 1A,B). At lower stimulation intensities (i.e., 10 or 20 APs), AP-2μ KO neurons displayed significantly attenuated exocytic responses (*Figure 4—figure supplement 1B*), possibly reflecting a reduced release probability originating from defects in SV reformation, akin to the reported phenotype of clathrin loss in hippocampal neurons (*Watanabe et al., 2013*; *Watanabe et al., 2014*).

Collectively, these data unravel a clathrin-independent role of the clathrin adaptor AP-2 in the endocytic retrieval of select SV cargos including VGLUT1 and VGAT at physiological temperature, while endocytosis of Syt1 or SV2A proceeds with unaltered kinetics in the absence of AP-2.

## CIE of endogenous VGAT depends on the clathrin adaptor AP-2

As optical imaging of SEP reporters may lead to artifacts caused by overexpression of exogenous SV proteins (*Opazo et al., 2010*), we analyzed the internalization kinetics of endogenous VGAT using antibodies directed against its luminal domain coupled to the pH-sensitive fluorophore CypHer5E. The cyanine-based dye CypHer5E is quenched at neutral pH but exhibits bright fluorescence when present in the acidic lumen of SVs (*Hua et al., 2011*) and, thus can serve as a tracer for the recycling of endogenous SV proteins when it is preloaded into SVs (e.g., by high-frequency stimulation or spontaneous labeling, see Materials and methods) prior to the measurements (*Figure 5A*). First, we probed the effects of AP-2μ KO on VGAT endocytosis. Loss of AP-2 severely delayed the endocytic retrieval of endogenous VGAT in response to train stimulation with either 200 APs (*Figure 5B, C*) or 50 APs (*Figure 5D, E*) at physiological temperature, consistent with our results from exogenously expressed VGAT-SEP (see *Figure 3*). To determine whether the requirement for AP-2 reflects a function for CME in the retrieval of endogenous VGAT, we examined the effects of genetic or pharmacological blockade of clathrin function. Lentiviral shRNA-mediated depletion of clathrin had no effect on the endocytic retrieval of endogenous VGAT in response to either strong (e.g., train of 200 APs applied at 40 Hz) (*Figure 5F, G*) or mild stimulation (50 APs at 20 Hz) (*Figure 5H, I*) at physiological temperature. Similar results were obtained, if clathrin function was perturbed pharmacologically by acute inhibition in the presence of Pitstop2 in either WT or clathrin-depleted neurons (*Figure 5B–I*). However, block of clathrin function potently inhibited CME of transferrin (*Figure 5J*) and resulted in a significant reduction of the readily releasable and total recycling vesicle pool sizes probed by

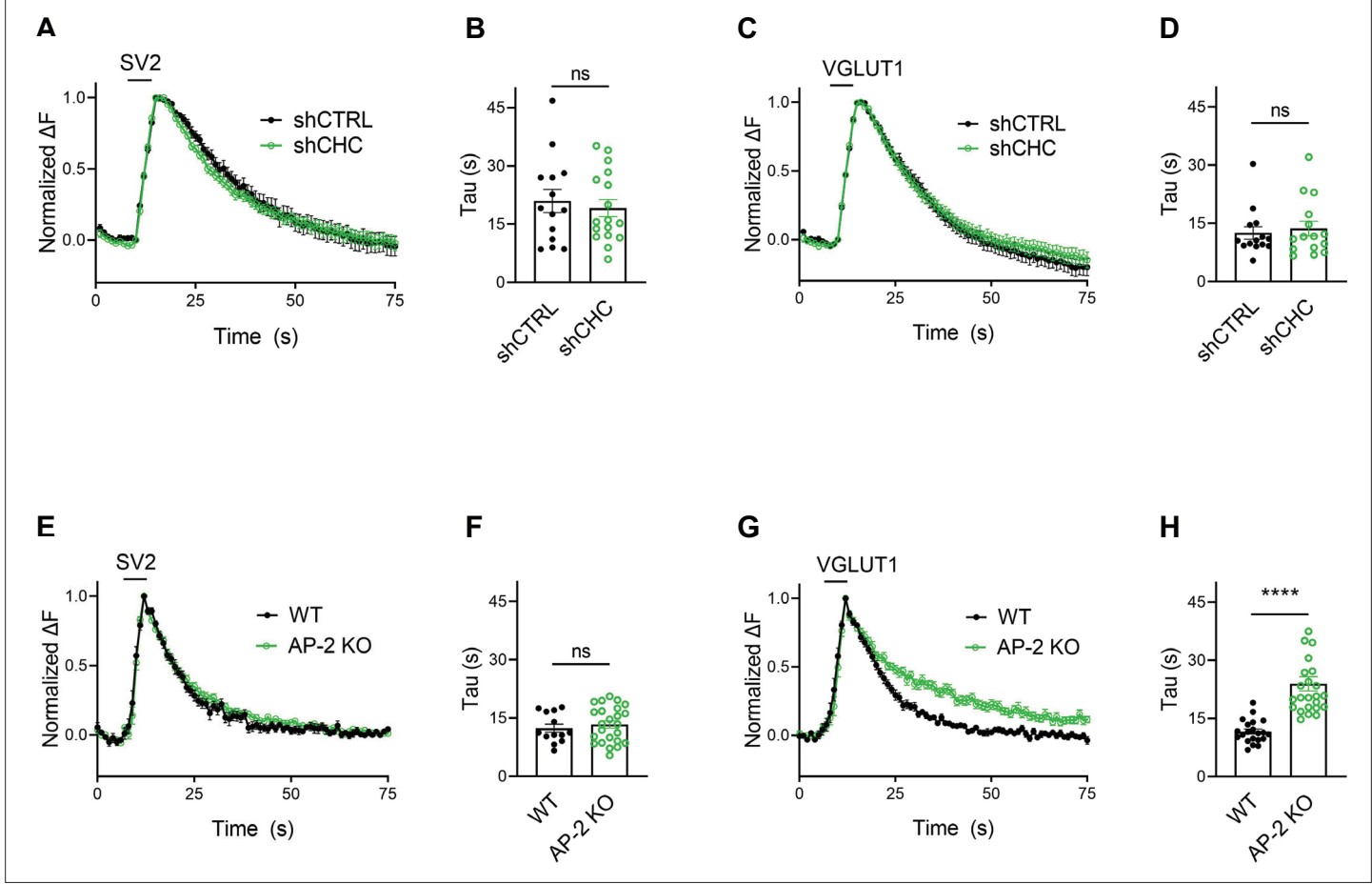

**Figure 4.** Clathrin-independent endocytic retrieval of synaptic vesicle (SV) proteins mediated by adaptor protein complex 2 (AP-2) is independent of the stimulation strength at physiological temperature. (**A–D**) Lack of clathrin does not alter the endocytosis of SV2 and VGLUT1 in response to stimulation with 50 APs (i.e., a stimulus that releases the RRP) at physiological temperature. Average normalized traces of neurons transduced with lentivirus expressing nonspecific shRNA (shCTRL) or shRNA-targeting CHC (shCHC) and cotransfected with either SEP-tagged SV2A (**A**) or VGLUT1 (**C**) stimulated with 50 APs applied at 20 Hz at physiological temperature. Quantification of the endocytic decay time constant ( τ ) in neurons coexpressing SV2A-SEP (**B**) and shCTRL (20.97 ± 2.99 s) or shCHC (19.18 ± 2.22 s); and VGLUT1-SEP (**D**) and shCTRL (12.51 ± 1.62 s) or shCHC (13.68 ± 1.90 s). Data represent the mean ± standard error of the mean (SEM) for SV2A ($n_{shCTRL}$ = 14 images, $n_{shCHC}$ = 17 images; p = 0.6274) and for VGLUT1 ($n_{shCTRL}$ = 14 images, $n_{shCHC}$ = 15 images; p = 0.6468). Two-sided unpaired $t$-test. (**E–H**) Endocytosis delay for VGLUT1 but not for SV2A in neurons depleted of AP-2 when stimulated with a mild train of 50 APs. Average normalized traces of neurons from WT and AP-2μ KO mice transfected with either SV2A-SEP (**E**) or VGLUT1-SEP (**G**) in response of 50 APs applied at 20 Hz at physiological temperature. Quantification of the endocytic decay time constant (τ) of SV2A-SEP-expressing neurons (**F**) ($\tau_{WT}$ = 12.40 ± 1.05 s, $\tau_{AP-2\mu\ KO}$ = 13.34 ± 0.95 s) or VGLUT1-SEP (**H**) ($\tau_{WT}$ = 11.61 ± 0.65 s, $\tau_{AP-2\mu\ KO}$ = 23.96 ± 1.87 s). Data represent the mean ± SEM for SV2A ($n_{WT}$ = 13 images, $n_{AP-2\mu\ KO}$ = 24 images; p = 0.5355) and for VGLUT1 ($n_{WT}$ = 21 images, $n_{AP-2\mu\ KO}$ = 24 images; ****p < 0.0001). Two-sided unpaired $t$-test. Raw data can be found in *Figure 4—source data 1*.

The online version of this article includes the following source data and figure supplement(s) for figure 4:

**Source data 1.** Source data for *Figure 4A-H*.

**Figure supplement 1.** Decreased fraction of active synapses in neurons lacking adaptor protein complex 2 (AP-2).

**Figure supplement 1—source data 1.** Source data for *Figure 4—figure supplement 1B*.

consecutive trains of 50 and 900 APs interspersed by a 90-s interstimulus interval (*Gerth et al., 2017*; *Figure 5K*). In contrast, the postexocytic retrieval kinetics of endogenous VGAT remained unaltered (*Figure 5L*).

We conclude that at physiological temperature the endocytosis of endogenous VGAT from the neuronal surface depends on the clathrin adaptor AP-2, while clathrin function is dispensable. Instead, clathrin may facilitate the reformation of functional SVs from ELVs downstream of CIE to sustain neurotransmission (*Kononenko et al., 2014*; *Watanabe et al., 2014*).

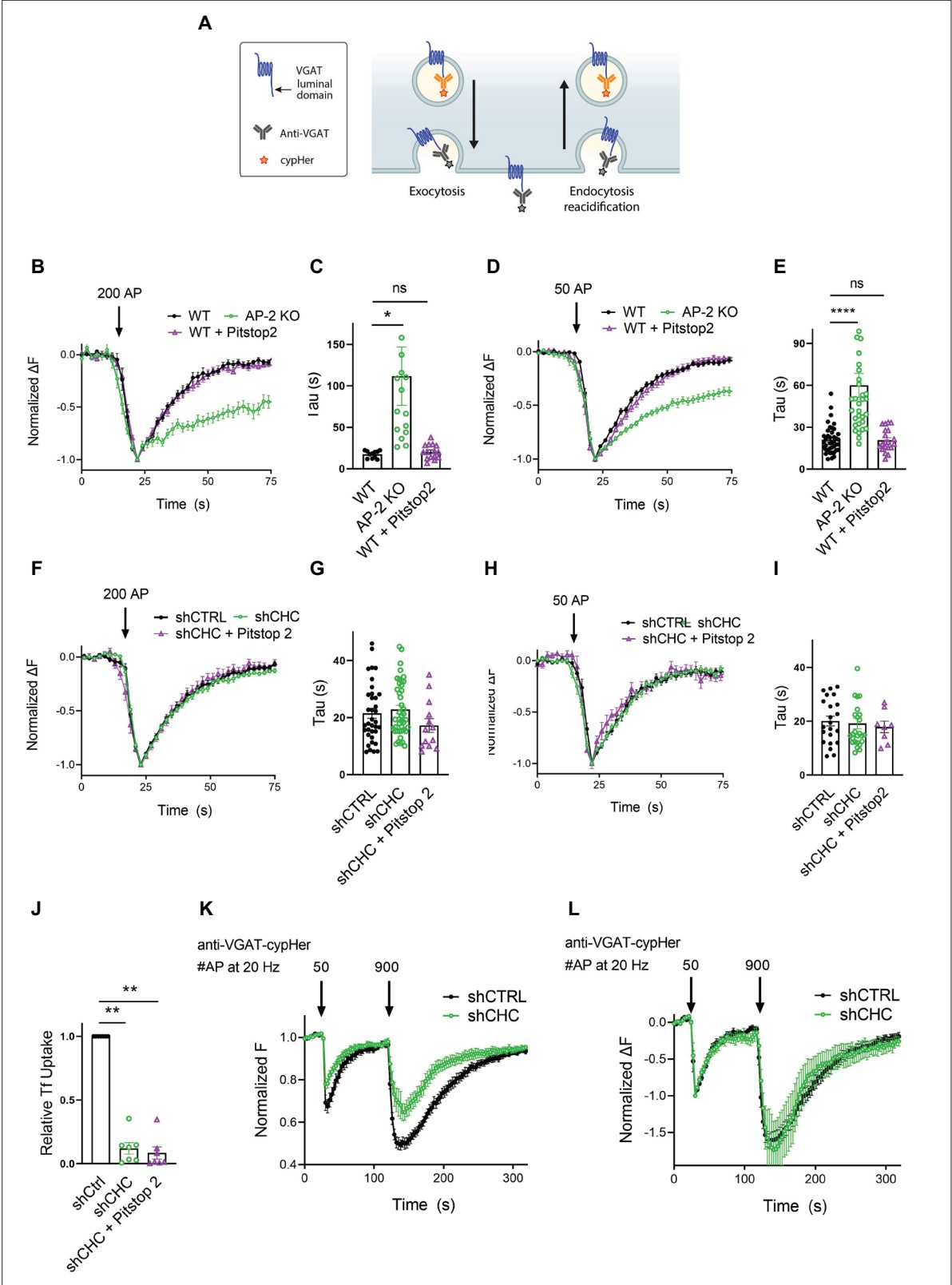

**Figure 5.** Postexocytic sorting of endogenous VGAT depends on adaptor protein complex 2 (AP-2) but not clathrin at physiological temperature. (**A**) Diagram depicting the use of CypHer5E-coupled antibodies targeting the luminal domain of VGAT to monitor fluorescence changes during exoendocytosis of endogenously labeled VGAT, as CypHer5E is a pH-sensitive fluorophore which is quenched at neutral extracellular pH. (**B–I**) Clathrin but not AP-2 is dispensable for endocytic retrieval of endogenous CypHer5E-labeled VGAT independent of the stimulation intensity at physiological

*Figure 5 continued on next page*

*Figure 5 continued*

temperature. Average normalized traces of neurons from WT treated or not with the clathrin inhibitor Pitstop2 and from AP-2μ KO mice incubated with anti-VGAT CypHer5E-coupled antibodies for live labeling of synapses in response to a high-frequency stimulus train (200 APs at 40 Hz) (**B**) or to mild-frequency stimulation (50 APs at 20 Hz) (**D**) at physiological temperature. (**C**) Quantification of the endocytic decay time constant ($\tau$) of anti-VGAT CypHer5E-labeled neurons stimulated with 200 APs at 40 Hz ($\tau_{WT}$ = 17.38 ± 1.30 s, $\tau_{AP-2\mu KO}$ = 111.7 ± 35.12 s, $\tau_{WT+Pitstop2}$ = 20.39 ± 2.23 s). Data represent the mean ± standard error of the mean (SEM) for $n_{WT}$ = 11 images, $n_{AP-2\mu KO}$ = 15 images, and $n_{WT+Pitstop2}$ = 15 images. *$p_{WT vs AP-2\mu KO}$ = 0.0179, $p_{WT vs WT+Pitstop2}$ = 0.9954, one-way analysis of variance (ANOVA) with Tukey's post-test. (**E**) Quantification of the endocytic decay constant ($\tau$) of anti-VGAT CypHer5E-labeled neurons after delivery of 50 APs at 20 Hz ($\tau_{WT}$ = 21.13 ± 1.57 s, $\tau_{AP-2\mu KO}$ = 60.04 ± 8.39 s, $\tau_{WT+Pitstop2}$ = 20.73 ± 1.81 s). Data represent the mean ± SEM for $n_{WT}$ = 40 images, $n_{AP-2\mu KO}$ = 32 images, and $n_{WT+Pitstop2}$ = 20 images. ****$p_{WT vs AP-2\mu KO}$ < 0.0001, $p_{WT vs WT+Pitstop2}$ = 0.9986, one-way ANOVA with Tukey's post-test. (**F**) Average normalized traces of neurons transduced with lentivirus expressing nonspecific shRNA (shCTRL) or shRNA-targeting CHC (shCHC) treated or not with the clathrin inhibitor Pitstop2 and incubated with anti-VGAT CypHer5E-coupled antibodies in response to a high-frequency stimulus train (200 APs at 40 Hz) (**F**) or to mild-frequency stimulation (50 APs at 20 Hz) (**H**) at physiological temperature. (**G**) Quantification of the endocytic decay time constant ($\tau$) of anti-VGAT CypHer5E-labeled neurons stimulated with 200 APs at 40 Hz ($\tau_{shCTRL}$ = 21.50 ± 1.72 s, $\tau_{shCHC}$ = 22.86 ± 1.46 s, $\tau_{shCHC+Pitstop2}$ = 17.23 ± 2.41 s). Data represent the mean ± SEM for $n_{shCTRL}$ = 36 images, $n_{shCHC}$ = 42 images, and $n_{shCHC+Pitstop2}$ = 13 images. $p_{shCTRL vs shCHC}$ = 0.8119, $p_{shCTRL vs shCHC+Pitstop2}$ = 0.3688, $p_{shCHC vs shCHC+Pitstop2}$ = 0.1683, one-way ANOVA with Tukey's post-test. (**I**) Quantification of the endocytic decay time constant ($\tau$) of anti-VGAT CypHer5E-labeled neurons after delivery of 50 APs at 20 Hz ($\tau_{shCTRL}$ = 20.09 ± 1.78 s, $\tau_{shCHC}$ = 19.23 ± 2.01 s, $\tau_{shCHC+Pitstop2}$ = 17.90 ± 2.15 s). Data represent the mean ± SEM for $n_{shCTRL}$ = 22 images, $n_{shCHC}$ = 26 images, and $n_{shCHC+Pitstop2}$ = 8 images. $p_{shCTRL vs shCHC}$ = 0.9427, $p_{shCTRL vs shCHC+Pitstop2}$ = 0.8286, $p_{shCHC vs shCHC+Pitstop2}$ = 0.9301, one-way ANOVA with Tukey's post-test. (**J**) Clathrin inactivation leads to reduced CME of transferrin. Quantification of primary neurons transduced with shCTRL or shCHC treated or not with the clathrin inhibitor Pitstop2 and allowed to internalize AlexaFluor[647]-labeled transferrin (Tf647) for 20 min at 37°C. Values for shCTRL were set to 1. The data represent mean ± SEM from $n$ = 7 independent experiments. ****$p$ < 0.0001, two-sided one-sample $t$-test. (**K, L**) Clathrin loss increases depression of neurotransmitter release without changing postexocytic retrieval kinetics of endogenous VGAT. (**K**) Average normalized traces of neurons transduced with lentivirus expressing either shCTRL or shCHC, incubated with anti-VGAT CypHer5E-coupled antibodies and subjected to consecutive stimulus trains of 50 and 900 APs applied both at 20 Hz with an interstimulus interval of 1.5 min to determine the size of the readily releasable synaptic vesicle (SV) pool and the recycling SV pool. $n$ = 10 images for shCTRL and $n$ = 8 images for shCHC. (**L**) Average normalized traces of neurons transduced with lentivirus expressing either shCTRL or shCHC, incubated with anti-VGAT CypHer5E-coupled antibodies and subjected to consecutive stimulus trains of 50 and 900 APs applied both at 20 Hz with an interstimulus interval of 1.5 min. The fluorescence was normalized to the first peak at the end of the first AP train with 50 APs. No differences in the kinetics of endocytic recovery were observed. $n$ = 10 images for shCTRL and $n$ = 8 images for shCHC. Raw data can be found in ***Figure 5—source data 1***.

The online version of this article includes the following source data for figure 5:

**Source data 1.** Souce data for ***Figure 5B-L***.

## AP-2 depletion causes surface stranding of endogenous vesicular neurotransmitter transporters but not of Syt1 and SV2A

As endocytosis of a subset of SV proteins, for example VGLUT1 and VGAT, was impaired in the absence of AP-2, one might expect their partial redistribution to the neuronal surface in AP-2μ KO neurons. To test this, we labeled surface-stranded SV proteins by a membrane-impermeant biotinylating reagent in cultured cerebellar granule neurons derived from AP-2μ KO mice or WT littermate controls. Biotinylated proteins were captured on a streptavidin-conjugated matrix and subsequently analyzed by immunoblotting (***Figure 6A***). No difference was detected in the plasma membrane levels of Syt1 and SV2A between WT and AP-2μ KO neurons (***Figure 6B–E***). By contrast, significantly larger amounts of VGLUT1 and VGAT were found at the plasma membrane of AP-2μ KO neurons compared to WT controls (***Figure 6F–I***), while the total levels of SV proteins assessed either by western blot or immunostaining were unaltered (***Figure 6—figure supplement 1A,B***).

To challenge these results by an independent approach, we took advantage of available antibodies that recognize the luminal domains of VGAT, Syt1, and VGLUT1 (***Figure 7***). Application of these antibodies under nonpermeabilizing conditions to selectively recognize the surface-stranded SV protein pool revealed elevated plasma membrane levels of VGAT (***Figure 7F–H***) and VGLUT1 (***Figure 7—figure supplement 1A-C***) in AP-2μ KO hippocampal neurons, while the presynaptic surface pool of Syt1 remained unaltered (***Figure 7B–D***). Importantly, silencing of neuronal activity in the presence of the sodium channel blocker tetrodotoxin (TTX) rescued surface stranding of VGAT (***Figure 7F–H***) and VGLUT1 (***Figure 7—figure supplement 1A-C***), suggesting that the observed plasma membrane accumulation of a subset of SV proteins in AP-2μ KO neurons is a consequence of defective stimulation-induced SV protein retrieval following exocytic SV fusion.

Collectively, these findings provide strong support for the hypothesis that the clathrin adaptor AP-2 is required for the endocytic retrieval of select SV cargos including VGLUT1 and VGAT under

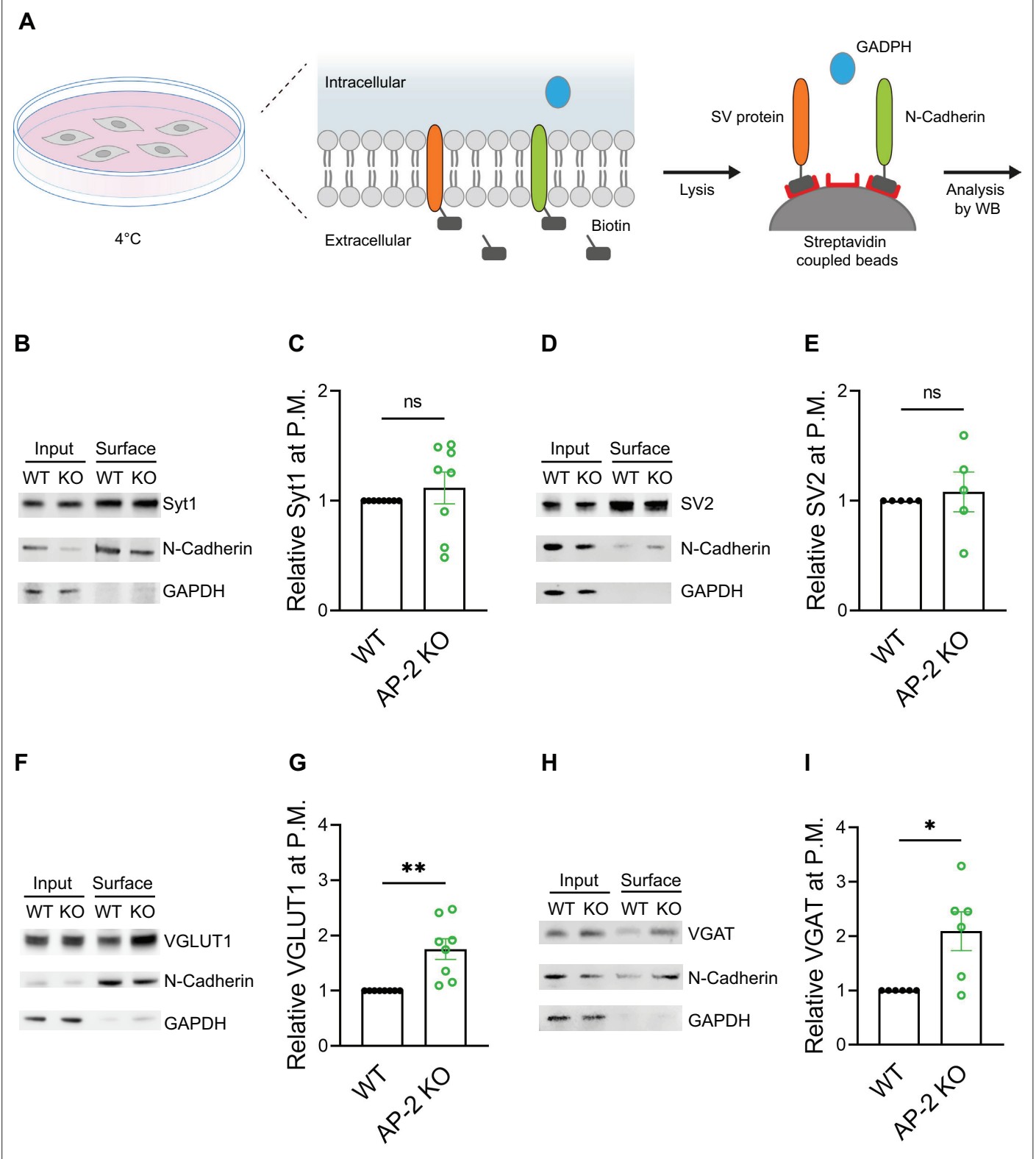

**Figure 6.** Adaptor protein complex 2 (AP-2) depletion results in surface stranding of endogenous vesicular neurotransmitter transporters but not of Synaptotagmin 1 and SV2A. (**A**) Schematic diagram of the workflow for cell surface protein enrichment. (**B–I**) AP-2 participates in the surface retrieval of the endogenous synaptic vesicle (SV) proteins such as VGLUT1 and VGAT but not of SV2 and Syt1. Cell surface proteins from WT and AP-2µ KO cerebellar granule cells were biotinylated and affinity-purified using streptavidin beads. Total (Input) and biotinylated proteins (Surface) were analyzed by western blot using antibodies against Syt1 (**B**), SV2 (**D**), VGLUT1 (**F**), and VGAT (**H**). N-cadherin and GAPDH were used as control of cell surface

*Figure 6 continued on next page*

*Figure 6 continued*

membrane and cytosol fraction, respectively. The fold surface enrichment of select proteins (**C, E, G, I**) in the absence of AP-2 was quantified. Values for WT neurons were set to 1. Data represent the mean ± standard error of the mean (SEM). $n_{Syt1} = 8$, $n_{SV2} = 5$, $n_{VGLUT1} = 8$, $n_{VGAT} = 6$ independent experiments. $p_{Syt1} = 0.4456$, $p_{SV2} = 0.6736$, $**p_{VGLUT1} = 0.0049$, $*p_{VGAT} = 0.0279$; two-sided one-sample *t*-test. Raw data can be found in *Figure 6—source data 1*, and *Figure 6—source data 2*.

The online version of this article includes the following source data and figure supplement(s) for figure 6:

**Source data 1.** Source data for *Figure 6C, E, G, I*.

**Source data 2.** Raw uncropped immunoblot images for *Figure 6B, D, F, H*.

**Figure supplement 1.** Adaptor protein complex 2 (AP-2) depletion does not change the total levels of synaptic vesicle (SV) proteins.

**Figure supplement 1—source data 1.** Source data for *Figure 6—figure supplement 1A, B*.

physiological conditions, thereby identifying a clathrin-independent function of AP-2 in the sorting of SV proteins at the presynaptic plasma membrane at central mammalian synapses.

## AP-2-binding deficient mutations in vesicular transporters phenocopy loss of AP-2

In a final set of experiments, we set out to determine the molecular basis for the clathrin-independent function of AP-2 in the sorting of SV proteins by focusing on VGLUT1 and VGAT. Previous studies had identified acidic cluster dileucine motifs (*Bonifacino and Traub, 2003*; *Kelly et al., 2008*) in the cytoplasmic tails of vesicular neurotransmitter transporters as putative interaction sites for AP-2 and possibly other clathrin adaptors (*Figure 8A*). As mutational inactivation of these motifs was further reported to delay the kinetics of VGLUT1 and VGAT analyzed at nonphysiological temperature (*Voglmaier et al., 2006*; *Santos et al., 2013*; *Li et al., 2017*), vesicular neurotransmitter transporters were proposed to be internalized via CME mediated by clathrin and AP-2 (*Mori and Takamori, 2017*). Given our data reported above, we hypothesized that these prior results might reflect the direct recognition of VGLUT1 and VGAT by AP-2 at the neuronal surface to enable their internalization via CIE at physiological temperature (see *Figure 1C*).

To probe this hypothesis, we first analyzed the association of the cytoplasmic C-terminal domain of VGLUT1 with the clathrin adaptor complex AP-2 and its close relatives AP-1 and AP-3. Robust binding of the GST-fused cytoplasmic domain of VGLUT1 to AP-2 was observed, whereas no association with AP-3 was detected (*Figure 8B, C*). Mutational inactivation of the putative AP-2-binding dileucine motif, that is $F_{510}A/V_{511}A$ (FV/AA) (*Voglmaier et al., 2006*; *Santos et al., 2013*; *Li et al., 2017*), largely abrogated VGLUT1 complex formation with AP-2. We also detected a weak, possibly nonspecific interaction of VGLUT1 with AP-1 that was insensitive to the FV/AA mutation (*Figure 8—figure supplement 1A*). These results show that VGLUT1 is directly recognized and binds to AP-2 via its acidic cluster dileucine motif.

To probe the functional significance of this interaction we monitored the endocytic retrieval of VGLUT1 and VGAT carrying mutations in their respective AP-2-binding dileucine motifs at physiological temperature. Mutant forms of VGLUT1 or VGAT defective in AP-2 binding displayed significantly slower endocytosis kinetics compared to the respective WT proteins (*Figure 8D–G*, wild type in black and VGLUT1-FV/AA mutant or VGAT-$F_{44}$A/AA mutant in purple). These endocytic defects were exacerbated when endocytosis was monitored at RT and under conditions that might favor CME (200 APs, 5 Hz; *Kononenko et al., 2014*; *Figure 8—figure supplement 1B-E*), consistent with earlier data (*Voglmaier et al., 2006*). Importantly, the delayed decay of mutant VGLUT1-FV/AA-SEP signals could not be attributed to defects in reacidification (e.g., caused by internalization into slowly acidifying compartments), because poststimulus application of acid solution effectively quenched its fluorescence (*Figure 8—figure supplement 1F,G*). To analyze whether the observed kinetic delay in the endocytosis of dileucine mutant VGLUT1 and VGAT variants was caused by loss of their ability to associate with AP-2, we monitored their retrieval in AP-2μ KO neurons. Strikingly, loss of AP-2 not only phenocopied the effect of mutational inactivation of the dileucine motifs in VGLUT1 or VGAT but combined mutational inactivation of the dileucine motifs in VGLUT1 or VGAT and AP-2μ KO did not result in additive phenotypes (*Figure 8D–G*).

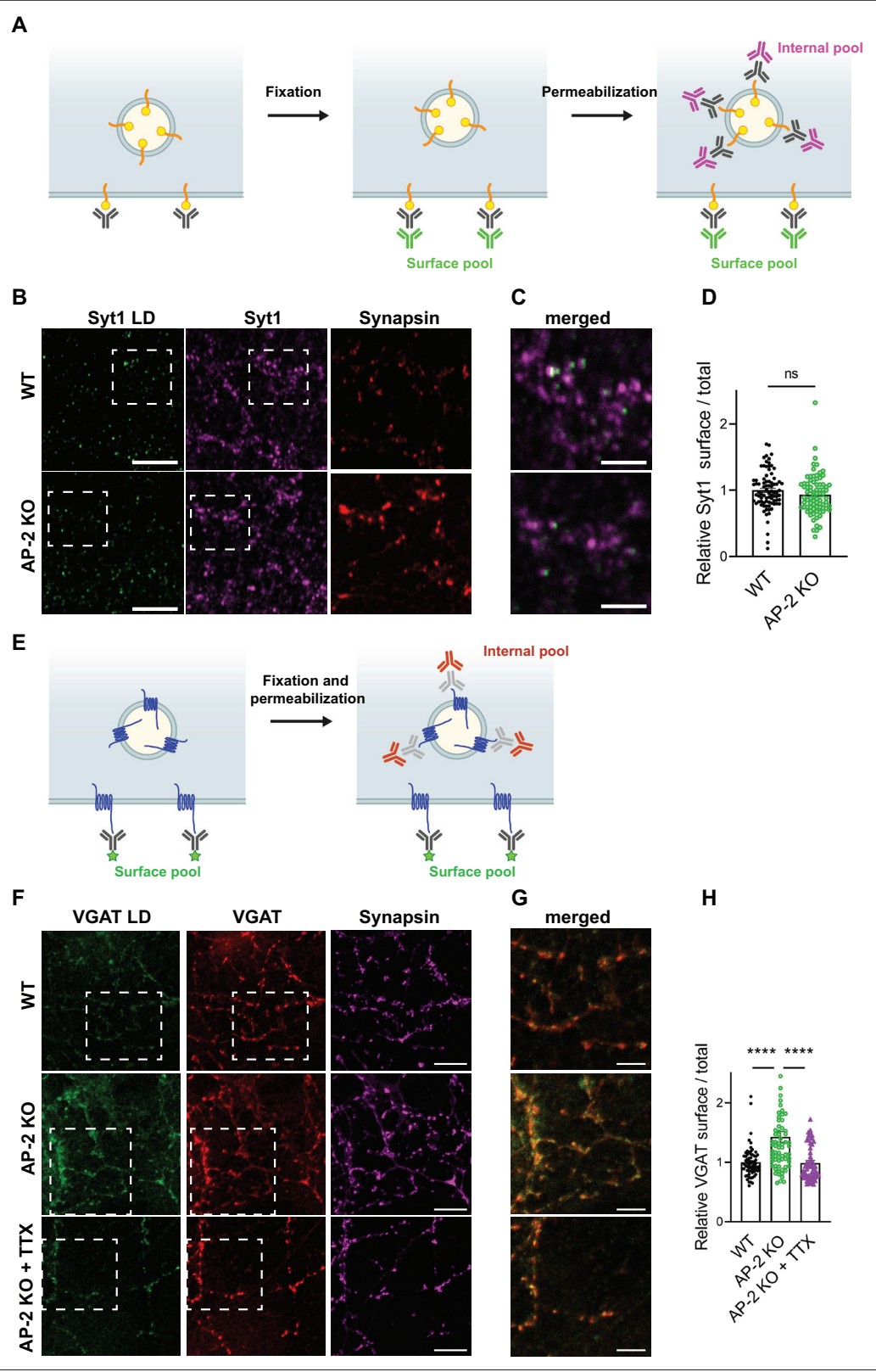

**Figure 7.** Surface stranding of adaptor protein complex 2 (AP-2)-dependent synaptic vesicle (SV) cargos in the absence of AP-2 is activity dependent. (**A–D**) Surface levels of Syt1 are unaffected by loss of AP-2. (**A**) Schematic drawing of the assay to monitor surface and total levels of Syt1. To label surface epitopes of Syt1, living hippocampal neurons are first incubated with anti-Syt1 antibodies against the luminal domain (black) at 4°C to limit

*Figure 7 continued on next page*

*Figure 7 continued*

its endocytosis prior to fixation. With no permeabilization conditions, neurons are incubated with 488-conjugated secondary antibodies (green) allowing to reveal the surface pool of Syt1. After washing off unbound antibodies, coverslips are subsequently immunostained using Syt1 antibodies against the cytosolic side (gray) after applying permeabilization conditions. Incubation with 647-conjugated secondary antibodies (magenta) will reveal the total amount of Syt1. Coverslips will be imaged to determine the amount of surface and total Syt1 labeling present in synapses by additional immunostaining of the presynaptic marker Synapsin (not depicted). (**B**) Representative confocal images of cultured hippocampal neurons from WT or AP-2μ KO mice coimmunostained for total Syt1 (magenta), surface Syt1 (Syt1 LD, green), and Synapsin (red). Scale bars, 5 μm. (**C**) A zoom of the marked area in (**B**). Scale bars, 2 μm. (**D**) Quantification of surface/total Syt1 levels. Values were normalized for WT. Data represent mean ± standard error of the mean (SEM) of $n_{WT}$ = 83 images and $n_{AP-2\mu\ KO}$ = 79 images. p = 0.1519, two-sided unpaired *t*-test. (**E–H**) Elevated surface levels of VGAT in the absence of AP-2 are rescued by blocking neuronal network activity. (**E**) Schematic drawing of the assay to monitor surface and total levels of VGAT. To label the surface pool of VGAT, living hippocampal neurons are first incubated with fluorophore-conjugated (green stars) antibodies (black) against the luminal domain of VGAT at 4°C prior to fixation. After permeabilization, coverslips are immunostained using VGAT antibodies against the cytosolic side (gray) and 568-conjugated secondary antibodies (orange) revealing the total VGAT. Coverslips are imaged for analyzing the surface and total VGAT labeling present in synapses by additional immunostaining of the presynaptic marker Synapsin (not depicted). (**F**) Representative confocal images of WT or AP-2μ KO hippocampal neurons treated or not with tetrodotoxin (TTX) since days in vitro (DIV)7 to block spontaneous action potentials and coimmunostained for total VGAT (red), surface VGAT (VGAT LD, green) and Synapsin (magenta). Scale bars, 10 μm. (**G**) A zoom of the marked area in (**F**). Scale bars, 5 μm. (**H**) Quantification shows that elevated ratio of surface/total VGAT in AP-2μ KO neurons is rescued when neurons were treated with TTX. Values were normalized to WT. Data represent mean ± SEM of $n_{WT}$ = 67 images, $n_{AP-2\mu\ KO}$ = 69 images, and $n_{AP-2\mu\ KO+TTX}$ = 58 images. ****$p_{WT\ vs\ AP-2\mu\ KO}$ < 0.0001, $p_{WT\ vs\ AP-2\mu\ KO+TTX}$ = 0.9904, ****$p_{AP-2\mu\ KO\ vs\ AP-2\mu\ KO+TTX}$ < 0.0001. One-way analysis of variance (ANOVA) with Tukey's post-test. Raw data can be found in *Figure 7—source data 1*.

The online version of this article includes the following source data and figure supplement(s) for figure 7:

**Source data 1.** Source data for *Figure 7D, H*.

**Figure supplement 1.** Adaptor protein complex 2 (AP-2) depletion alters the localization of VGLUT1 in an activity-dependent manner.

**Figure supplement 1—source data 1.** Source data for *Figure 7—figure supplement 1C*.

These data show that AP-2 directly recognizes surface-stranded VGLUT1 and VGAT via acidic cluster dileucine motifs contained in their cytoplasmic domains to facilitate their endocytic retrieval from the plasma membrane via CIE.

## Discussion

Our findings based on lentiviral depletion of clathrin and conditional KO of AP-2 in hippocampal neurons reveal a crucial clathrin-independent function of the clathrin adaptor AP-2 in the endocytic sorting of a subset of SV proteins at central synapses. Several lines of evidence support this view: First, comprehensive survey of the endocytic retrieval of six major SV proteins by optical imaging conducted in two independent laboratories provides strong support for the emerging notion (*Soykan et al., 2017*; *Watanabe et al., 2014*) that SV endocytosis occurs independent of clathrin, corroborating the prevalence of CIE at physiological temperature. Second and most surprisingly, we find that the endocytic retrieval of a subset of these SV proteins including VGLUT1 and VGAT from the presynaptic plasma membrane depends on sorting by the clathrin adaptor AP-2. This conclusion from SEP- and CypHer5E-based imaging and acid quenching experiments of exogenously expressed or endogenous SV proteins is further corroborated by the observation that a fraction of endogenous VGLUT1 and VGAT molecules remain stranded on the presynaptic plasma membrane of AP-2μ KO neurons, a phenotype that is rescued upon silencing of neuronal activity. Impaired endocytosis of VGLUT1 in the absence of AP-2 may indirectly also impact on the kinetics of Synaptophysin and Synaptobrevin/VAMP2 retrieval (*Figure 3I–L*), consistent with the proposed function of VGLUT1 as a master orchestrator of the retrieval of a subset of SV proteins (*Pan et al., 2015*). However, other possibilities such as a direct or indirect association of Synaptophysin and/or Synaptobrevin/VAMP2 with AP-2, cannot be ruled out. Finally, we show that AP-2-mediated efficient sorting of VGLUT1 and VGAT during CIE

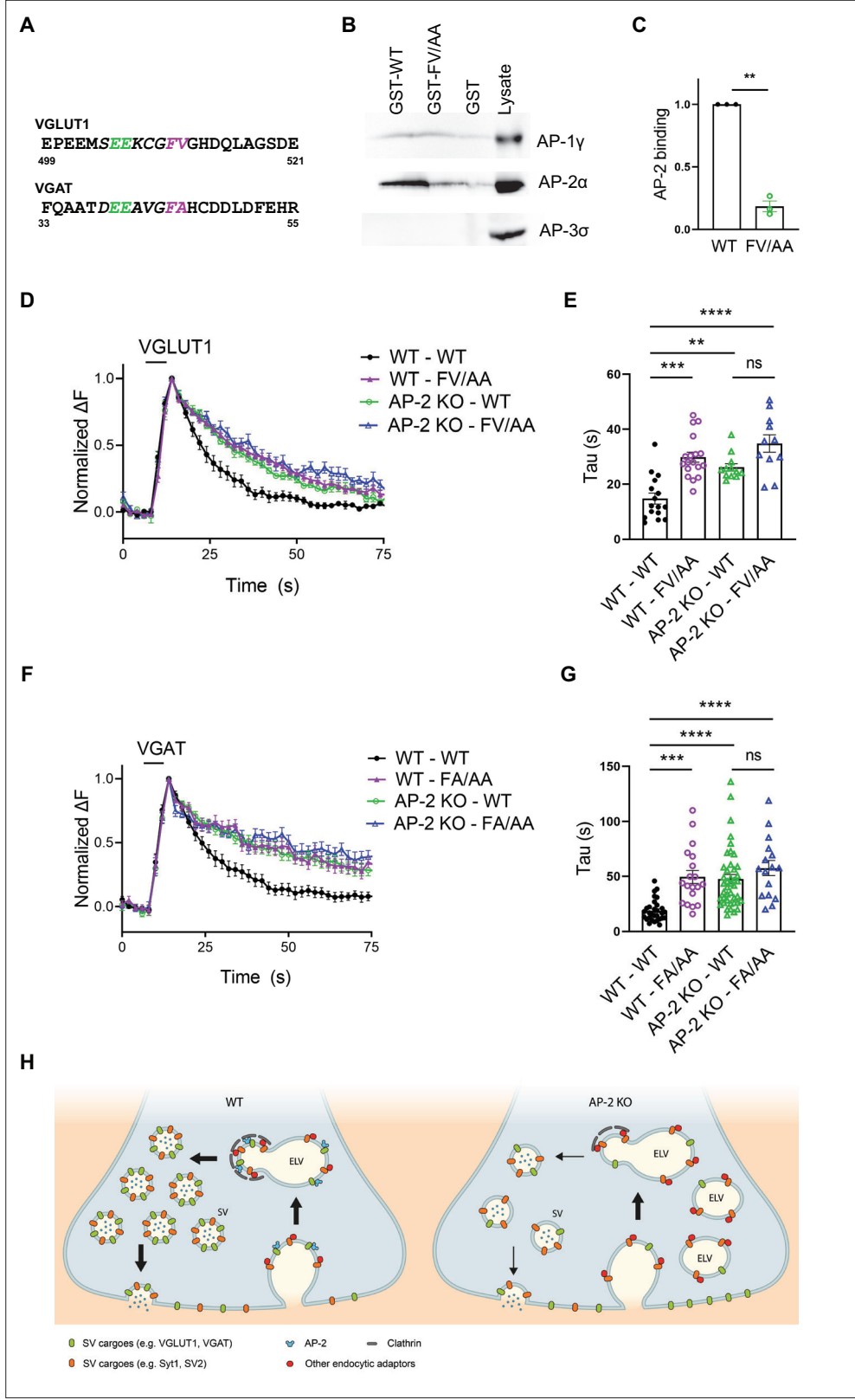

**Figure 8.** Adaptor protein complex 2 (AP-2)-binding deficient mutants of vesicular neurotransmitter transporters phenocopy loss of AP-2. (**A–C**) Association of the cytoplasmic domain of VGLUT1 with the clathrin adaptor complex AP-2 is abolished upon mutational inactivation of the putative AP-2-binding dileucine motif, that is $F_{510}A/V_{511}A$ (FV/AA). (**A**) Acidic cluster dileucine-like motifs identified in the C-terminal cytoplasmic tail of mouse VGLUT1

*Figure 8 continued on next page*

*Figure 8 continued*

and in the N-terminal cytoplasmic tail of mouse VGAT. Numbers indicate amino acid numbers of the respective proteins. Green and magenta indicate two acidic amino acids and two hydrophobic amino acids conserved within the motifs, respectively. (**B**) Immunoblot analysis of material affinity-purified via GST-VGLUT1 C-terminus-WT, GST-VGLUT1 C-terminus-FV/AA or GST alone, and brain lysate using specific antibodies against AP-2, AP-1, and AP-3 shows that dileucine-like motif found in the C-terminus of VGLUT1 binds preferentially to AP-2. (**C**) Quantified data exhibit VGLUT1 C-terminus-FV/AA variant to significantly disrupt interaction with AP-2. Data represent the mean ± standard error of the mean (SEM) from $n = 3$ independent experiments. **p = 0.0027, two-sided one-sample *t*-test. (**D–G**) Mutant variants of VGLUT1 or VGAT defective in AP-2-binding display significantly slower endocytosis kinetics in response to stimulation in a similar manner to be observed in the absence of AP-2. (**D**) Average normalized traces of neurons from WT and AP-2μ KO mice transfected with either the WT or the mutant variant (FV/AA) of VGLUT1-SEP in response of a stimulus train of 200 APs applied at 40 Hz at physiological temperature. (**E**) Quantification of the endocytic decay time constant ($\tau$) of VGLUT1-SEP-expressing neurons ($\tau_{WT\text{-}WT} = 14.82 \pm 1.98$ s, $\tau_{WT\text{-}FV/AA} = 29.82 \pm 1.82$ s, $\tau_{AP\text{-}2\mu\ KO\text{-}WT} = 26.15 \pm 1.34$ s, $\tau_{AP\text{-}2\mu\ KO\text{-}FV/AA} = 34.80 \pm 3.18$ s). Data represent the mean ± SEM of $n_{WT\text{-}WT} = 16$ images, $n_{WT\text{-}FV/AA} = 18$ images, $n_{AP\text{-}2\mu\ KO\text{-}WT} = 12$ images, and $n_{AP\text{-}2\mu\ KO\text{-}FV/AA} = 11$ images. ****p$_{WT\text{-}WT\ vs\ WT\text{-}FV/AA} < 0.0001$, **p$_{WT\text{-}WT\ vs\ AP\text{-}2\mu\ KO\text{-}WT} = 0.0023$, ****p$_{WT\text{-}WT\ vs\ AP\text{-}2\mu\ KO\text{-}FV/AA} < 0.0001$, p$_{AP\text{-}2\mu\ KO\text{-}WT\ vs\ AP\text{-}2\mu\ KO\text{-}FV/AA} = 0.0530$, one-way analysis of variance (ANOVA) with Tukey's post-test. (**F**) Average normalized traces of WT and AP-2μ KO neurons transfected with either the WT or the mutant variant (FA/AA) of VGAT-SEP in response of 200 APs applied at 40 Hz at physiological temperature. (**G**) Quantification of the endocytic decay time constant ($\tau$) of VGAT-SEP-expressing neurons ($\tau_{WT\text{-}WT} = 18.86 \pm 1.79$ s, $\tau_{WT\text{-}FA/AA} = 49.56 \pm 5.95$ s, $\tau_{AP\text{-}2\mu\ KO\text{-}WT} = 47.43 \pm 4.19$ s, $\tau_{AP\text{-}2\mu\ KO\text{-}FA/AA} = 57.60 \pm 6.92$ s). Data represent the mean ± SEM of $n_{WT\text{-}WT} = 30$ images, $n_{WT\text{-}FA/AA} = 19$ images, $n_{AP\text{-}2\mu\ KO\text{-}WT} = 42$ images, and $n_{AP\text{-}2\mu\ KO\text{-}FA/AA} = 16$ images. ***p$_{WT\text{-}WT\ vs\ WT\text{-}FA/AA} = 0.0001$, ****p$_{WT\text{-}WT\ vs\ AP\text{-}2\mu\ KO\text{-}WT} < 0.0001$, ****p$_{WT\text{-}WT\ vs\ AP\text{-}2\mu\ KO\text{-}FA/AA} < 0.0001$, p$_{AP\text{-}2\mu\ KO\text{-}WT\ vs\ AP\text{-}2\mu\ KO\text{-}FA/AA} = 0.4561$, one-way ANOVA with Tukey's post-test. (**H**) Illustrated model proposing a clathrin-independent role for dedicated endocytic adaptors such as AP-2 which recognize select exocytosed synaptic vesicle (SV) proteins (e.g., VGLUT1 and VGAT) present on the neuronal surface to facilitate their clathrin-independent endocytic internalization while clathrin operates downstream facilitating the reformation of functional SVs by budding from internal endosome-like vacuoles (ELVs) in a process that also depends on AP-2 and other clathrin-associated endocytic proteins. Raw data can be found in *Figure 8—source data 1* and *Figure 8—source data 2*.

The online version of this article includes the following source data and figure supplement(s) for figure 8:

**Source data 1.** Source data for *Figure 8C-G*.

**Source data 2.** Raw uncropped immunoblot images for *Figure 8B*.

**Figure supplement 1.** Defective retrieval of vesicular neurotransmitter transporters carrying mutations in the adaptor protein complex 2 (AP-2)-binding dileucine motif cannot be ascribed to defects in acidification.

**Figure supplement 1—source data 1.** Source data for *Figure 8—figure supplement 1A-G*.

---

is achieved by the recognition of acidic cluster dileucine motifs by AP-2, in agreement with earlier biochemical and cell biological experiments (*Voglmaier et al., 2006*; *Santos et al., 2013*; *Li et al., 2017*). Our data thus underscore the importance of AP-2-mediated sorting of select SV cargo during CIE, in the absence of which the compositional integrity of SVs becomes perturbed. This mechanism may also be of pathological relevance in humans. For example, defective endocytosis of VGAT and resulting defects in inhibitory neurotransmission may underlie developmental and epileptic enceph-alopathy caused by a pathogenic loss-of-function variant of AP-2μ in human patients (*Helbig et al., 2019*).

Our findings are most consistent with and support a mechanism of SV recycling in which dedi-cated endocytic adaptors such as AP-2 and others (e.g., AP180, Stonin 2) operate at the presynaptic cell surface where they recognize and recruit SV proteins to sites of CIE. How AP-2 is targeted to presynaptic endocytic sites is unclear, but likely involves coincident detection of phosphatidylinositol 4,5-bisphosphate, a lipid selectively enriched at the plasma membrane, and cargo proteins (e.g., VGLUT1 or VGAT), possibly in conjunction with other endocytic factors. Clathrin assembly only occurs once endocytic vesicles have pinched off from the plasma membrane, that is downstream of CIE, to reform functional SVs by budding from internal ELVs. This process of SV reformation depends on AP-2 and other clathrin-associated endocytic proteins (*Figure 8H*). Such an integrated model, in which AP-2 acts at two sites, endocytic sites on the presynaptic surface and on internal ELVs together with clathrin, not only explains previous observations pertaining to the speed of SV endocytosis (*Delvendahl et al., 2016*; *Watanabe et al., 2013*) and the apparent lack of effect of clathrin loss

on SV membrane internalization in various models (*Kononenko et al., 2014*; *Soykan et al., 2017*; *Kasprowicz et al., 2008*; *Heerssen et al., 2008*), but is also consistent with the slow kinetics of clathrin assembly (*McMahon and Boucrot, 2011*) and the accumulation of SV proteins on the neuronal surface in the absence of dedicated endocytic adaptors for SV proteins, for example Stonin 2, AP180/ CALM (*Kononenko and Haucke, 2015*; *Cousin, 2017*; *Kaempf et al., 2015*; *Koo et al., 2015*; *Mori and Takamori, 2017*), and AP-2 (this study). We speculate that the mechanism identified here also operates during UFE. Interestingly, recent quantitative proteomic analysis of rodent brain has revealed AP-2 but not clathrin to be highly enriched on SVs (*Taoufiq et al., 2020*), suggesting that AP-2 may interact with SV cargos prior to the *bona fide* endocytic process. This perpetual interaction of AP-2 with SV cargos might thus enable rapid sorting and endocytic internalization, that is during UFE or other forms of CIE. Of note, our findings are also consistent with recent data regarding a calcium-independent form of SV endocytosis that appears to operate independent of clathrin (*Orlando et al., 2019*).

An important question raised by our work is how AP-2-independent SV cargos such as Synaptotagmin 1 and SV2 are endocytically retrieved from the neuronal surface. While endocytic adaptors for SV2 have not been reported, studies in mouse hippocampal neurons (*Kononenko et al., 2013*; *Lee et al., 2019*) and in invertebrate models (*Stimson et al., 2001*; *Phillips et al., 2010*; *Jung et al., 2007*) have identified cargo-selective roles of Stonin 2 and the related SGIP1 protein in the endocytic retrieval of Synaptotagmin 1 from the presynaptic cell surface. Enigmatically however, it was shown that while loss of Stonin 2 causes the partial accumulation of Synaptotagmin 1 at plasma membrane sites near the AZ, the kinetics of SV endocytosis appeared to be even accelerated (*Kaempf et al., 2015*; *Kononenko et al., 2013*). One possibility therefore is that Synaptotagmin 1 due to its comparably large presynaptic surface fraction (*Pan et al., 2015*) does not require active endocytic sorting. Instead, it may reach sites of endocytosis either by lateral diffusion or via confinement near sites of exocytic release (*Hua et al., 2011*; *Kononenko et al., 2013*; *Willig et al., 2006*). SV2 could follow Synaptotagmin 1 via a piggy-back mechanism (*Haucke and De Camilli, 1999*), consistent with the finding that both proteins can form a stable complex in vivo (*Kaempf et al., 2015*; *Lazzell et al., 2004*; *Schivell et al., 1996*; *Yao et al., 2010*). A confinement-based mechanism may also support the endocytic retrieval of other SV proteins in the absence of their specific sorting adaptors (e.g., VGLUT1 or VGAT endocytosis in AP-2μ KO neurons). Future studies are needed to address this possibility in more detail.

From a more general perspective our findings dissent from the widely held view that AP-2 obligatorily associates with clathrin to execute its cell physiological functions, at least in central nervous system (CNS) neurons. While the most well-known function of AP-2 is its involvement in CME in mammalian cells and tissues, studies in higher fungi have uncovered a clathrin-independent role of fungal AP-2 in the polar localization of the lipid flippases DnfA and DnfB (*Martzoukou et al., 2017*). Interestingly, in this system AP-2 is seen to colocalize with endocytic markers and the actin-associated protein AbpA, but not with clathrin (*Martzoukou et al., 2017*). Because AP-2 also colocalizes with a fungal homolog of Synaptobrevin (*Martzoukou et al., 2017*), and clathrin-independent SV endocytosis at hippocampal synapses depends on actin polymerization (*Soykan et al., 2017*), our newly observed function of AP-2 might reflect an unexpectedly widely conserved endocytic mechanism. Conversely, studies in AP-2 KO cells have revealed AP-2-independent forms of CME in mammals that impact on receptor sorting and signaling (*Pascolutti et al., 2019*).

Taken together our findings together with other studies suggest an unexpected plasticity of endocytic mechanisms in eukaryotes including the mammalian CNS.

## Materials and methods

**Key resources table**

| Reagent type (species) or resource | Designation | Source or reference | Identifiers | Additional information |
|---|---|---|---|---|
| Genetic reagent (*M. musculus*) | ICR wild type | SLC Japan | RRID:MGI:5462094 | |

*Continued on next page*

*Continued*

| Reagent type (species) or resource | Designation | Source or reference | Identifiers | Additional information |
|---|---|---|---|---|
| Genetic reagent (*M. musculus*) | C57BL/6N wild type | Leibniz Research Institute for Molecular Pharmacology | RRID:MGI:5651595 | |
| Genetic reagent (*M. musculus*) | *Ap2m1*<sup>lox/lox</sup> × inducible CAG-Cre | Haucke Lab *Kononenko et al., 2014* | | |
| Recombinant DNA reagent | SEP-tagged VGLUT1 | Takamori Lab *Mori et al., 2021* | | |
| Recombinant DNA reagent | SEP-tagged Syp | L. Lagnado *Granseth et al., 2006* | | |
| Recombinant DNA reagent | SEP-tagged Syp | Takamori Lab *Egashira et al., 2015* | | |
| Recombinant DNA reagent | SEP-tagged Syb2 | S. Kawaguchi | | |
| Recombinant DNA reagent | pLenti6PW-STB | Y. Fukazawa *Hioki et al., 2009* | | |
| Recombinant DNA reagent | SEP-tagged Syt1 | J. Klingauf *Wienisch and Klingauf, 2006* | | |
| Recombinant DNA reagent | SEP-tagged Syb2 | J. Klingauf *Wienisch and Klingauf, 2006* | | |
| Recombinant DNA reagent | SEP-tagged SV2A | E.R. Chapman *Kwon and Chapman, 2011* | | |
| Recombinant DNA reagent | SEP-tagged VGLUT1 | R. Edwards and S. Voglmaier *Voglmaier et al., 2006* | | |
| Recombinant DNA reagent | SEP-tagged VGAT | S. Voglmaier *Santos et al., 2013* | | |
| Recombinant DNA reagent | f(U6)sNLS-RFPw msClathrin scrambled | C. Rosenmund *Watanabe et al., 2014* | | |
| Recombinant DNA reagent | f(U6)sNLS-RFPw msClathrin shRNA | C. Rosenmund *Watanabe et al., 2014* | | |
| Recombinant DNA reagent | pLV-shmCHC-hPGK-mCherry | VectorBuilder | | |
| Recombinant DNA reagent | AP2mu-IRES-mRFP in-AAV-HBA-EWB | Haucke Lab *López-Hernández et al., 2020* | | |
| Chemical compound, drug | Pitstop2 | Haucke Lab *von Kleist et al., 2011* | | 30µM |
| Chemical compound, drug | Folimycin | Sigma | Cat# C9705 | 67nM |
| Chemical compound, drug | EZ-Link Sulfo-NHS-LC-Biotin | Thermo Fisher Scientific | Cat# 21,335 | 0.5mg/ml |
| Chemical compound, drug | Streptavidin agarose | Millipore | Cat# 69,203 | 100µl |
| Chemical compound, drug | GST-bind resin | Cytiva | Cat# 17075601 | 120µl |
| Chemical compound, drug | Transferrin 647 | Life Technologies | Cat# T23366 | 25µg/ml |
| Chemical compound, drug | Tamoxifen ((*Z*)-4-hydroxytamoxifen) | Sigma | Cat# H7904 | 0.1µM |
| Chemical compound, drug | AraC | Sigma | Cat# C6645 | 4µM |
| Chemical compound, drug | Uridine | Sigma | Cat# U3750 | 100µM |
| Chemical compound, drug | Uridine | Sigma | Cat# U3003 | 100µM |
| Chemical compound, drug | FUDR | Sigma | Cat# F0503 | 40µM |

*Continued*

| Reagent type (species) or resource | Designation | Source or reference | Identifiers | Additional information |
|---|---|---|---|---|
| Commercial assay or kit | ProFection Mammalian Transfection System – Calcium Phosphate | Promega | Cat# E1200 | |
| Commercial assay or kit | CalPhos Mammalian Transfection Kit | Takara | Cat# 631,312 | |
| Antibody | AP-2 α-adaptin (mouse monoclonal) | Haucke Lab | AP6 | IF: 1:100 |
| Antibody | AP-2 α-adaptin (mouse monoclonal) | Santa Cruz | Cat# sc-55497 RRID:AB_2056344 | WB: 1:100 |
| Antibody | γ-Adaptin 1 (mouse monoclonal) | BD Biosciences | Cat# 610,386 RRID:AB_397769 | WB: 1:500 |
| Antibody | σ-Adaptin 3 (mouse monoclonal) | Haucke Lab | SA4 | WB: 1:250 |
| Antibody | β-Actin (mouse monoclonal) | Sigma-Aldrich | Cat# A-5441 RRID:AB_476744 | WB: 1:5000 |
| Antibody | GAPDH (mouse monoclonal) | Sigma-Aldrich | Cat# G8795 RRID:AB_1078991 | WB: 1:5000 |
| Antibody | N-cadherin (mouse monoclonal) | BD Biosciences | Cat# 610,920 RRID:AB_610920 | WB: 1:1000 |
| Antibody | CHC (mouse monoclonal) | Haucke lab | TD1 | WB: 1:500 |
| Antibody | CHC (rabbit polyclonal) | Abcam | Cat# ab21679 RRID:AB_ 2083165 | IF: 1:1000 |
| Antibody | Synaptophysin (mouse monoclonal) | R. Jahn | Cl 7.2 | IF: 1:1000 |
| Antibody | MAP-2 (guinea pig polyclonal) | Synaptic Systems | Cat# 188 004 RRID:AB_2138181 | IF: 1:400 |
| Antibody | Synapsin (mouse monoclonal) | Synaptic Systems | Cat# 106 001 RRID:AB_887805 | IF: 1:400 |
| Antibody | Synapsin (guinea pig polyclonal) | Synaptic Systems | Cat# 106 004 RRID:AB_1106784 | IF: 1:400 |
| Antibody | VGLUT1 (guinea pig polyclonal) | Synaptic Systems | Cat# 135 304 RRID:AB_887878 | IF: 1:300 WB: 1:500 |
| Antibody | Synaptotagmin 1 (rabbit polyclonal) | Synaptic Systems | Cat# 105 102 RRID:AB_887835 | IF: 1:100 |
| Antibody | Synaptotagmin 1 (mouse monoclonal) | Synaptic Systems | Cat# 105 011 RRID:AB_887832 | IF: 1:250 WB: 1:500 |

| Reagent type (species) or resource | Designation | Source or reference | Identifiers | Additional information |
|---|---|---|---|---|
| Antibody | VGAT Oyster 488 (rabbit polyclonal) | Synaptic Systems | Cat# 131 103C2 RRID:AB_10640329 | IF: 1:100 |
| Antibody | VGAT Oyster 568 (rabbit polyclonal) | Synaptic Systems | Cat# 131 103C3 RRID:AB_887867 | IF: 1:100 |
| Antibody | VGAT (rabbit polyclonal) | Synaptic Systems | Cat# 131 013 RRID:AB_2189938 | IF: 1:300 |
| Antibody | VGAT (guinea pig polyclonal) | Synaptic Systems | Cat# 131 004 RRID:AB_887873 | WB: 1:500 |

*Continued*

| Reagent type (species) or resource | Designation | Source or reference | Identifiers | Additional information |
|---|---|---|---|---|
| Antibody | VGAT (rabbit polyclonal) | Synaptic Systems | Cat# 131 103CpH RRID:AB_2189809 | Live imaging:1:120 |
| Antibody | SV2 (rabbit polyclonal) | Abcam | Cat# ab32942 RRID:AB_778192 | WB: 1:500 |
| Antibody | Peroxidase-AffiniPure Anti-Rabbit IgG (H + L) (goat polyclonal) | Jackson ImmunoResearch Labs | Cat# 111-035-003 RRID:AB_2313567 | WB: 1:10,000 |
| Antibody | Peroxidase-AffiniPure Anti-Mouse IgG (H + L) (goat polyclonal) | Jackson ImmunoResearch Labs | Cat# 115-035-003 RRID:AB_10015289 | WB: 1:10,000 |
| Antibody | Peroxidase-AffiniPure Anti-guineapig IgG (H + L) (goat polyclonal) | Jackson ImmunoResearch Labs | Cat# 106-035-003 RRID:AB_2337402 | WB: 1:10,000 |
| Antibody | Mouse IgG HRP Linked F(ab')2 Fragment (sheep monoclonal) | Cytiva | Cat# GENA9310 RRID:AB_772193 | WB: 1:10,000 |
| Antibody | Anti-mouse IgG, IRDye 800CW Conjugated (goat polyclonal) | LI-COR Biosciences | Cat# 926-32210 RRID:AB_621842 | WB: 1:10,000 |
| Antibody | IRDye 680RD anti-mouse IgG (H + L) (goat polyclonal) | LI-COR Biosciences | Cat# 925-68070 RRID:AB_2651128 | WB: 1:10,000 |
| Antibody | IRDye 680RD anti-rabbit IgG (H + L) (goat polyclonal) | LI-COR Biosciences | Cat# 926-68071 RRID:AB_10956166 | WB: 1:10,000 |
| Antibody | Anti-rabbit IgG, IRDye 800CW Conjugated (goat polyclonal) | LI-COR Biosciences | Cat# 926-32211 RRID:AB_621843 | WB: 1:10,000 |
| Antibody | Anti-guinea pig IgG Alexa Fluor 647 (goat polyclonal) | Thermo Fisher Scientific | Cat# A-21450 RRID:AB_141882 | IF: 1:500 |
| Antibody | Anti-guinea pig IgG Alexa Fluor 488 (donkey polyclonal) | Jackson ImmunoResearch Labs | Cat# 706-545-148 RRID:AB_141954 | IF: 1:500 |
| Antibody | Anti-guinea pig IgG Alexa Fluor 568 (goat polyclonal) | Thermo Fisher Scientific | Cat# A-11075 RRID:AB_141954 | IF: 1:500 |
| Antibody | Anti-mouse IgG Alexa Fluor 568 (goat polyclonal) | Thermo Fisher Scientific | Cat# A-11004 RRID:AB_2534072 | IF: 1:500 |
| Antibody | Anti-rabbit IgG Alexa Fluor 647 (goat polyclonal) | Thermo Fisher Scientific | Cat# A-21245 RRID:AB_2535813 | IF: 1:500 and 1:1000 |
| Antibody | Anti-rabbit IgG Alexa Fluor 488 | Thermo Fisher Scientific | Cat# A-11008 RRID:AB_143165 | IF: 1:500 |
| Antibody | Anti-mouse IgG Alexa Fluor 488 (goat polyclonal) | Thermo Fisher Scientific | Cat# A-11029 RRID:AB_2534088 | IF: 1:500 and 1:1000 |
| Software and algorithms | Image J | Open Source Software | https://imagej.net/Welcome | |
| Software and algorithms | Prism v.8 | GraphPad | RRID: SCR_002798 | |
| Software and algorithms | Python | Programming Language | RRID: SCR_008394 | |

## Animals

Primary neurons for the experiments presented in *Figures 2 and 4A–D*, and in *Figure 2—figure supplement 1* and *Figure 8—figure supplement 1* were obtained from ICR mice. Pregnant ICR mice were purchased from SLC, Japan.

Primary neurons for the experiments presented in *Figures 3–8* and the rest of figure supplements were obtained from either wild-type C57BL/6 or conditional AP-2µ KO (*Ap2m1$^{lox/lox}$* × inducible CAG-Cre) mice previously described (*Kononenko et al., 2014*). All mice were given food and water ad

libitum. Animals were kept in a local animal facility with a 12 hr light and 12 hr dark cycle. Ambient temperature was maintained around 21°C with a relative humidity of 50%. The health reports can be provided upon request. Mice from both genders were used for experiments. Littermates were randomly assigned to experimental groups. Multiple independent experiments were carried out using several biological replicates specified in the figure legends.

## Cell line

For production of lentiviral vectors, tsA201 cell line was used as a host. The origin of the cell line is human embryonic kidney and stably expressing an SV40 temperature-sensitive T antigen. The cells have been eradicated from mycoplasma at ECACC, and the identity of tsA201 has been confirmed by STR profiling.

## Preparation of neuronal cell cultures

Primary hippocampal cultures for the experiments performed in *Figures 2 and 4A–D*, *Figure 2—figure supplement 1*, and *Figure 8—figure supplement 1* were prepared from embryonic day 16 ICR mice as described previously (*Egashira et al., 2015*; *Egashira et al., 2016*), with slight modifications. Briefly, hippocampi were dissected, and incubated with papain (90 units/ml, Worthington) for 20 min at 37°C. After digestion, hippocampal cells were plated onto poly-D-lysine-coated coverslips framed in a Nunc 4-well dish (Thermo Fisher) at a cell density of 20,000–30,000 cells/cm$^2$ and grown in Neurobasal medium (Thermo Fisher) supplemented with 1.25% FBS, 2% B27, and 0.5 mM glutamine at 37°C, 5% $CO_2$. On 2–3 days in vitro (DIV), 40 µM FUDR (Sigma) and 100 µM uridine (Sigma) were added to the culture medium to limit glial proliferation. One-fifth of the culture medium was routinely replaced with fresh neurobasal medium supplemented with 2% B27 and 0.5 mM glutamine every 2–4 days.

To prepare primary hippocampal and cerebellar neurons for the experiments presented in *Figures 3–8* and the figure supplement 1 for *Figures 2–4* and *Figure 6*, hippocampus or cerebellum were surgically removed from postnatal mice at P1–3 or P6, respectively. This was followed by trypsin digestion and dissociation into single neurons. Primary neurons were plated onto poly-L-lysine-coated coverslips for 6-well plates and cultured in Modified Eagle Medium (MEM) medium (Thermo Fisher) containing 2% B27% and 5% FCS. The medium for cerebellar cultures additionally contained 25 mM KCl. To avoid astrocyte growth, hippocampal cultures were treated with 2 µM AraC. To deplete AP-2µ subunit, cultured neurons from floxed conditional AP-2µ KO mice expressing a tamoxifen-inducible Cre recombinase were treated with 0.25 µM (Z)-4-hydroxytamoxifen (Sigma) at DIV3. Neurons derived from floxed littermates that were Cre negative were used as controls and treated with equal amounts of (Z)–4-hydroxytamoxifen.

## Plasmids

SEP-tagged Syt1 (NM_001252341.1), VGLUT1 (NM_182993.2), and SV2A (NM_057210.2) were designed as previously described (*Voglmaier et al., 2006*; *Diril et al., 2006*; *Kwon and Chapman, 2011*), and generated by In-Fusion recombination (Takara Bio). VGAT-SEP was constructed by fusing SEP to the luminal C-terminus of VGAT (NM_031782.2), preceded by GAATCC via In-Fusion recombination. Syp-SEP and Syb2-SEP were kind gifts from L. Lagnado (Sussex, UK) and S. Kawaguchi (Kyoto, Japan), respectively (*Granseth et al., 2006*; *Kawaguchi and Sakaba, 2015*). All SEP-tagged constructs were cloned in pcDNA3.1 expression vector or pLenti6PW lentiviral expression vector carrying a TRE promoter (*Hioki et al., 2009*). pLenti6PW lentiviral expression vector containing a human synapsin I promoter that drives a neuron-specific expression of advanced tetracycline transactivator (tTAad) was a generous gift from Y. Fukazawa (Fukui, Japan), and used to induce a protein expression under the control of TRE promoter (*Hioki et al., 2009*). VGLUT1-F$_{510}$A/V$_{511}$A and VGAT-F$_{44}$A/AA were made by PCR mutagenesis. Cytoplasmic C-terminal region of VGLUT1 (a.a. 496–560) was determined using Expasy ProtScale (https://web.expasy.org/protscale/) and subcloned into pGEX6P1 vector via *Bam*HI and *Sal*I sites. For CHC knockdown experiments in *Figure 2*, U6-promoter-based lentiviral shRNA vectors targeting mouse CHC (5′-GTTGGTGACCGTTGTTATG-3′) (*Watanabe et al., 2014*) or luciferase (5′-CCTAAGGTTAAGTCGCCCTCG-3′) as a nonsilencing control (*Cai et al., 2006*) were obtained from VectorBuilder biotechnology Co. Ltd (Kanagawa, Japan). To identify transduced cells, the shRNA vectors contained a mCherry sequence downstream of a hPGK promoter sequence.

For the data presented in *Figures 3, 4 and 8*, plasmids encoding for Syb2 and Syt1 (*Wienisch and Klingauf, 2006*) with a TEV protease cleavable SEP-tag were a kind gift from J. Klingauf (University of Münster, Münster, Germany). Syp-SEP was a kind gift from L. Lagnado (Sussex, UK). VGLUT1-SEP-tag was a kind gift from R. Edwards and S. Voglmaier (UCSF, CA, USA). SV2A-SEP (*Kwon and Chapman, 2012*) was a kind gift from E.R. Chapman (UW-Madison, WI, USA). VGAT-SEP was a gift from S. Voglmaier (Addgene plasmid #78578; http://n2t.net/addgene:78578; RRID:Addgene_78578). For the clathrin knockdown experiments performed in *Figure 5*, expression vectors f(U6)sNLS-RFPw msClathrin scrambled and f(U6)sNLS-RFPw msClathrin shRNA were a kind gift from C. Rosenmund (Berlin, Germany) (*Kononenko et al., 2014*; *Watanabe et al., 2014*). For rescue experiments shown in *Figure 3*, we used a construct previously described (*López-Hernández et al., 2020*) containing murine untagged AP-2µ followed by an IRES site by mRFP in an adenoviral AAV-HBA-EWB vector backbone, which serves to identify transfected neurons. An mRFP-expressing vector lacking AP-2µ cDNA was used as a control.

## Antibodies

### Immunoblotting
Secondary antibodies were all species specific. Horseradish peroxidase (HRP)-conjugated or LI-COR 800CW and 680RD infrared suitable antibodies were applied at 1:10,000 in blocking solution. Quantification was done based on chemiluminescence or fluorescence using an Odyssey FC detection system. Each panel of a figure has individual antibodies shown at the same exposure settings throughout the experiment.

### Immunofluorescence
Secondary antibodies were all species specific. Secondary antibodies fluorescently labeled with Alexa dyes 488, 568, or 647 (Thermo Fisher Scientific) were applied at 1:1000 or 1:500 in blocking solution. Antibodies used in this study are listed in Key Resource Table.

## Drug application
Pitstop2 (Leibniz-Forschungsinstitut für Molekulare Pharmakologie, Berlin, Germany) was used to inhibit clathrin-dependent endocytosis (*von Kleist et al., 2011*). Growth media of primary hippocampal neurons at DIV13–15 were replaced by osmolarity-adjusted, serum-free NBA medium (Gibco) for 1 hr prior the incubation with 30 µM of Pitstop2 during 1 hr. TTX (Sigma) was used to inhibit voltage-gated sodium channels and silence synaptic activity in cultured hippocampal neurons. Where indicated, neurons were incubated with 1 µM TTX at DIV7, which was renewed on DIV11.

## Transfection of primary neurons
Calcium phosphate transfection was carried out as previously described (*Jiang and Chen, 2006*), with slight modifications, using CalPhos Mammalian Transfection Kit (Takara Bio or Promega) on DIV7. Shortly, 6 µg plasmid DNA and 248 mM $CaCl_2$ dissolved in water were mixed with equal volume of 2× 2-[4-(2-Hydroxyethyl)-1-piperazinyl]ethanesulfonic acid (HEPES) buffered saline (total of 100 µl for one 35 mm dish), and incubated for 15–20 min allowing for precipitate formation, while neurons were starved in osmolarity-adjusted, serum-free MEM (Sigma) or NBA medium (Gibco) for the same time at 37°C, 5% $CO_2$. Precipitates were added to neurons and incubated at 37°C, 5% $CO_2$ for 30–40 min. Finally, neurons were washed by incubation in fresh MEM or osmolarity-adjusted Hanks' Balanced Salt Solution (HBSS) (Gibco) at 37°C, 10% $CO_2$ for 15 min and transferred back into their conditioned medium. For rescue experiments AP-2µ-IRES-RFP construct was introduced at DIV7 and the neurons were analyzed at DIV14.

## Lentivirus transduction of primary neurons
For experiments presented in *Figures 2 and 4A–D*, *Figure 2—figure supplement 1*, and *Figure 8—figure supplement 1*, lentivirus was produced using tsA201 cell line as described previously (*Egashira et al., 2015*; *Egashira et al., 2016*). The cells were transduced with lentivirus expressing VGLUT1-SEP and its mutant on DIV2, Syt1-SEP, Syp-SEP, Syb2-SEP, VGAT-SEP, and its mutant on DIV5–7, and shRNA for CHC on DIV7. To activate protein expression under the control of TRE promoter, lentivirus expressing tTAad was cotransduced on the same DIV. Transduction of clathrin shRNA on earlier DIV

caused severe loss of neurons at the time of recordings, and thus, cells were cotransduced with clathrin shRNA on DIV7 and experiments were conducted on DIV14. For clathrin knockdown experiments, lentiviral particles were prepared as follows: HEK293T cells were cotransfected with the lentivirus shuttle vector (10 µg) and two helper plasmids, pCMVdR8.9 and pVSV.G (5 µg each) using the calcium phosphate method. After 48 and 72 hr, virus-containing supernatant was collected, filtered, aliquoted, snap frozen in liquid nitrogen and stored at −80°C. Viruses were titrated with mice WT hippocampal mass-cultured neurons using NLS-RFP signals. For the clathrin knockdown experiments performed in *Figure 5*, mouse hippocampal neurons were transduced at DIV2, resulting in CHC depletion at DIV14 from the start of the treatment.

## Live imaging

For *Figures 2 and 4A–D*, *Figure 2—figure supplement 1*, and *Figure 8—figure supplement 1*, fluorescence imaging was performed on IX71 inverted microscope (Olympus) equipped with a ×60 (1.35 NA) oil-immersion objective and 75 W xenon arc lamp (Ushio). Cells on coverslips were mounted on a custom-made imaging chamber equipped on a movable stage with constant perfusion of Tyrode's solution (140 mM NaCl, 2.4 mM KCl, 10 mM HEPES, 10 mM glucose, 2 mM CaCl$_2$, 1 mM MgCl$_2$, 0.02 mM CNQX, 0.025 mM D-APV, adjusted to pH 7.4). Temperature was clamped at physiological temperature (35 ± 2°C) or RT (25 ± 2°C) using TC-324C temperature controller (Warner Instruments) with feedback control, SH-27B inline solution heater (Warner Instruments) and a custom-equipped air-heater throughout the experiment. Electrical field stimulation was delivered via bipolar platinum electrodes with 1-ms constant voltage pulses (50 V) controlled by pCLAMP Software (Molecular Devices). Fluorescence images (512 × 512 pixels) were acquired with ORCA-Flash 4.0 sCMOS camera (Hamamatsu Photonics) in time-lapse mode either at 1 fps (for imaging in response to 200-AP stimulation) or 2 fps (for imaging in response to 50-APs stimulation) under the control of MetaMorph software (Molecular Devices). SEP fluorescence was imaged with 470/22 nm excitation and 514/30 nm emission filters, and mCherry fluorescence with 556/20 nm excitation and 600/25 nm emission filters.

For SEP-based assays presented in *Figures 3, 4E–H and 8*, *Figure 3—figure supplement 1*, and *Figure 4—figure supplement 1*, cultured neurons at DIV13-15 were placed into an RC-47FSLP stimulation chamber (Warner Instruments) for electrical field stimulation and imaged at 37°C in osmolarity-adjusted basic imaging buffer (170 mM NaCl, 3.5 mM KCl, 20 mM N-Tris[hydroxy-methyl]-methyl-2-aminoethane-sulphonic acid [TES], 0.4 mM KH$_2$PO$_4$, 5 mM glucose, 5 mM NaHCO$_3$, 1.2 mM MgCl$_2$, 1.2 mMNa$_2$SO$_4$, 1.3 mM CaCl$_2$, 50 mM AP5, and 10 mM CNQX, pH 7.4) by epifluorescence microscopy (Nikon Eclipse Ti by MicroManager 4.11, eGFP filter set F36-526, and a sCMOS camera [Neo, Andor] equipped with a ×40 oil-immersion objective). For evaluation of general exo-/endocytosis, neurons were stimulated with 200 or 50 APs (40 or 20 Hz, respectively, 100 mA) and imaged at 1 or 0.5 fps with 100 ms excitation at 488 nm. For quantification of active synapses, signals >4× SD of the noise were considered as the threshold to identify ROIs that show stimulus-dependent changes in SEP-fluorescence signals following stimulation.

To distinguish between defective endocytosis and impaired reacidification of SVs carrying SEP-tagged SV proteins, acid quench assays were performed. Neurons were locally perfused with acidic imaging buffer in which TES was replaced by 2-(N-morpholino) ethane-sulfonic acid (MES), adjusted to pH 5.5 for 30 s before and after electrical stimulation. To obtain $\Delta F_1$, fluorescence values after acidic buffer treatment were subtracted from the initial fluorescence signals in neutral imaging buffer. $\Delta F_2$ was calculated by subtraction of the fluorescence values after acidic buffer treatment from the average fluorescence signals in neutral imaging buffer after the second acid quench poststimulation.

The fraction of acid-resistant VGLUT1-SEP molecules at 30-s poststimulation was determined essentially as described previously (*Soykan et al., 2017*). Briefly, extracellular VGLUT1-SEP fluorescence was first quenched for 15 s by locally perfusing acidic imaging buffer following a brief wash with neutral buffer. A second acid pulse of 10-s duration was applied at 30 s after the end of the stimulus to quench all VGLUT1-SEP fluorescence on the surface. The difference between the fluorescent signals during the first and the second acid wash corresponds to the fraction of VGLUT1-SEP retrieved from the plasma membrane during the stimulus and the subsequent 30-s chase period. This value was divided by the peak amplitude of VGLUT1-SEP fluorescence following stimulation to calculate the fraction of VGLUT1 internalized before the second acid wash.

To monitor recycling of endogenous VGAT, hippocampal neurons were labeled with CypHer5E-conjugated antibodies directed against the luminal domain of VGAT (#131,103CpH; Synaptic Systems; 1:120 from a 1 mg/ml stock) by incubation in their own conditioned culture medium at 37°C for 1 hr to allow antibody uptake and labeling of spontaneously active boutons. Neurons were washed with imaging buffer and placed in the stimulation chamber for electrical field stimulation and live imaging was performed as described above for SEP-based assays. Images were acquired at 1 fps with 100ms excitation at 647 nm. For *Figure 5K, L*, time-lapse mode was done at 1 frame every 3 s.

## Image and data analysis

Quantitative analysis of responding boutons was performed in Fiji (*Schindelin et al., 2012*) using Time Series Analyzer plugin (https://imagej.nih.gov/ij/plugins/time-series.html) or by using custom-written macros (available at https://github.com/DennisVoll/pHluorin_ROI_selector/, *Lopez-Henandez, 2021* copy archived at swh:1:rev:35b2db1fa1102ad2fa8b9131c588752766272c41). Circular regions of interest (ROIs, 4 µm² area) were manually positioned at the center of fluorescent puncta that appear stable throughout all trials and responded to an electrical stimulus, and the fluorescence was measured over time. Another five ROIs of the same size were positioned at the regions where no cell structures were visible, and their average fluorescence was subtracted as background signals. After further subtracting base signals, the fluorescence of each time point was normalized with the peak value. Time constant of endocytosis (Tau) was determined by fitting monoexponential decay curve [0 + A × exp(−x/tau)] using 'scipy.optimize.curve_fit' function in Python (*Virtanen et al., 2020*) or Prism 8 (Graphpad) softwares. Data of <30 boutons from a single experiment were averaged and counted as *n* = 1 for Tau calculation. All data were collected from two to five independent preparations.

## Photobleaching correction

Decrease in the fluorescence intensity signals due to photobleaching was corrected as previously described (*Hua et al., 2011*). The decay constant $\tau$ due to photobleaching was obtained from observations of intensity-time courses of nonactive boutons by experimentally fitting a monoexponential decay curve as follows: $I(t) = A \times \exp(−t/\tau)$, with $I(t)$, as fluorescence intensity at time $t$; $A$, as initial intensity $I(0)$; and $\tau$, as time constant of photobleaching. To correct for photobleaching, the fluorescence intensity value of the photobleaching function calculated at given time $t$ was summed up to the mean fluorescence intensity for every time point of the recording.

## Immunocytochemical analysis of cultured neurons

For the experiments presented in *Figure 2—figure supplement 1*, cultured hippocampal neurons transduced with nonsilencing shRNA or shRNA-targeting mouse CHC were fixed on DIV14 with 4% (wt/vol) paraformaldehyde and 4% sucrose in phosphate-buffered saline (PBS) for 15 min at RT. After washing in PBS, fixed cells were permeabilized with 0.2% Triton X-100 in PBS for 10–15 min and blocked in PBS containing 10% (vol/vol) FBS for 30 min at RT. Cells were then incubated with rabbit anti-CHC (1:1000, abcam, ab21679) and mouse anti-synaptophysin (1:1000, a kind gift from R. Jahn [Göttingen, Germany]) antibodies for 2 hr at RT, and subsequently, with anti-rabbit IgG Alexa Fluor 647 (1:1000, Thermo Fisher, A-21245) and anti-mouse IgG Alexa Fluor 488 (1:1000, Thermo Fisher, A-11029) antibodies for 45 min at RT. Transduced cells, visible by mCherry expression, were imaged using the same microscope setup with live imaging. Synaptophysin signals were imaged with 482.5/12.5 nm excitation and 530/20 nm emission filters, CHC signals with 628/20 nm excitation and 692/20 nm emission filters, and mCherry fluorescence with 540/10 nm excitation and 575IF-emission filters. Ratiometric quantification of CHC signals over synaptophysin signals was conducted in the automated fashion using Fiji with a custom-written macro as previously described (*Watanabe et al., 2013*), with slight modifications. In short, acquired images were first background subtracted using 'Rolling Ball' function with the radius set at 30 pixels (http://fiji.sc/Rolling_Ball_Background_Subtraction). Then, the synapses are defined by thresholding the synaptophysin signals using built-in 'Default' method (https://imagej.net/Auto_Threshold.html#Default). The binary image of synaptophysin was used as the regions of interest. The average intensities of CHC and synaptophysin were measured from those locations and were divided to obtain ratio between those two proteins. For experiments presented in *Figure 3—figure supplement 1* and *Figure 6—figure supplement 1*, primary hippocampal neurons seeded on coverslips were fixed for 13 min with 4% PFA in PBS solution on ice and washed three times

with PBS. Cells were permeabilized and blocked in blocking solution (PBS, 10% goat serum and 0.1% Triton X-100) for 30 min and incubated with primary antibodies diluted in blocking solution for 1 hr. After three washes with PBS, coverslips were incubated for 1 hr with secondary antibodies diluted in blocking solution, followed by three washes in PBS. Coverslips were mounted in Immu-Mount (Thermo Fisher) with 1.5 mg/ml 4,6-diamidino-2-phenylindole (DAPI; Sigma) to stain nuclei and were visualized routinely using the Zeiss laser scanning confocal microscope LSM710.

To distinguish between surface and internal SV protein pool, hippocampal neurons (DIV13–15) were gently washed once using osmolarity-adjusted HBS (25 mM HEPES, 140 mM NaCl, 5 mM KCl, 1.8 mM CaCl$_2$, 0.8 mM MgCl$_2$, 10 mM glucose, pH 7.4) prior to live labeling surface-localized VGAT, Syt1, or VGLUT1 with specific antibodies against their luminal/extracellular regions (VGAT Oyster 488: SySy, #131103C2 or VGAT Oyster 568, SySy, #131103C3; Synaptotagmin 1: SySy, #105102; VGLUT1: SySy, #135304) for 20 min at 4°C. After washing twice, cells were fixed for 5 min at RT in 4% PFA and washed again three times for 3 min. Without permeabilization, neurons were stained with Alexa Fluor 488 secondary antibody (except for VGAT which was previously incubated with an Oyster-labeled antibodies) for 45 min at RT, to visualize the surface pool of the corresponding SV protein. For labeling the internal population of those SV proteins in the same experiment, neurons were permeabilized for 10 min using 0.1% Triton X-100 and subsequently coverslips were incubated for 1 hr at RT with primary antibodies recognizing the cytosolic side of Synaptotamin 1 (SySy, #105011), VGAT (SySy, #131013), and VGLUT1 (SySy, #135304); and stained with Alexa Fluor 647 or 568 for 45 min at RT revealing the internal fraction of the SV protein pool. After four final washing steps of 3 min, the coverslips were mounted in Immu-Mount with DAPI to stain nuclei. Samples were visualized using the Zeiss laser scanning confocal microscope LSM710 using a ×63 oil objective. All acquisition settings were set equally for all groups within each experiment. Surface and internal fluorescent intensities were individually quantified using ImageJ and the ratio between such surface and internal signals was calculated for WT and KO conditions.

For quantifying the levels of presynaptic proteins (VGAT, VGLUT1, and Syt1) in axons, Synapsin staining signals (SySy, #106004 or #106011) were used as a mask to restrict the quantified are to the shape of synapsin-positive boutons by applying thresholding using ImageJ. Values were normalized to WT.

## Transferrin uptake

Primary neurons expressing lentivirally delivered CHC-targeting shRNA or a scramble version (DIV14) were starved for 1 hr in osmolarity-adjusted NBA medium (Gibco) at 37°C, 5% CO$_2$ and treated with 25 µg/ml transferrin coupled to Alexa Fluor 647 (Tf-647, Life technologies) in NBA medium for 20 min at 37°C, 5% CO$_2$. To remove unbound Tf-647, neurons were washed twice with cold PBS, followed by 1 min of acid-wash at pH 5.3 (cold 0.1 M acetic acid supplemented with 0.2 M NaCl) to quench surface bound Tf-647 and finally twice with cold PBS prior to 30 min fixation at RT with 4% (wt/vol) paraformaldehyde (PFA) and 4% sucrose in PBS. Coverslips were mounted in Immu-Mount with DAPI to stain nuclei and were visualized routinely using the Zeiss laser scanning confocal microscope LSM710 and fluorescence intensities per cell were quantified using ImageJ. First, a threshold was set to extract the punctate signal from cytosolic background and Tf-647 fluorescence intensities were then normalized to the corrected total blue channel fluorescence.

## Protein expression and purification

GST-fusion proteins were expressed in *Escherichia coli* BL21 cells overnight at 25°C after induction at OD$_{600}$ 0.4–0.7 with 1 mM isopropyl β-D-1-thiogalactopyranoside. Cells were harvested, resuspended in sonication buffer (50 mM Tris–Cl pH 8.0, 50 mM NaCl, 1 mM EDTA), lysed by ultrasonication followed by 1% Triton X-100 treatment and spun at 39,191 × $g$ for 30 min at 4°C. Proteins were then purified from the supernatant using Glutathione Sepharose 4B resin (Cytiva), according to the manufacturer's instructions, and dialyzed against pulldown buffer (20 mM HEPES, pH 7.4, 140 mM NaCl, 1 mM EDTA, 1 mM DTT).

## GST pulldown assay

Extracts from mouse brain were solubilized in pulldown buffer containing cOmplete Protease Inhibitor Cocktail (Roche) and 0.1% saponin for 45 min at 4°C, sedimented at 20,400 × $g$ for 25 min at 4°C, and

the supernatant (~16 mg total protein) was incubated with 600 µg of GST fusion proteins immobilized on Glutathione Sepharose for 160 min at 4°C. After pelleting, the beads were washed and bound protein was detected by immunoblot analysis using mouse monoclonal anti-γ-adaptin 1 (BD Biosciences, # 610386), anti-α-adaptin 2 (Santa Cruz, # sc-55497), and anti-σ-adaptin 3 (Haucke Lab, SA4) antibodies as primary antibodies. Fiji was used to quantify the intensity of bands.

## Cell surface biotinylation assay

Primary cerebellar granule neurons at DIV10 were treated as previously described (*López-Hernández et al., 2020*). Briefly, neurons were placed on ice, washed twice with ice-cold PBS$^{2+}$ (137 mM NaCl, 2.7 mM KCl, 8.1 mM Na$_2$HPO$_4$, 0.5 mM CaCl$_2$, 1 mM MgCl$_2$, pH 7.4) and incubated with 0.5 mg/ml sulfo-NHS-LC-biotin (EZ-Link, Pierce/Thermo Scientific) in PBS$^{2+}$ while shaking for 20 min at 4°C. The biotinylation solution was removed, and surplus biotin was quenched by two 5 min washes with 50 mM glycine in PBS$^{2+}$ at 4°C on a shaker. Cells were then washed briefly with PBS and scraped into lysis buffer (20 mM HEPES, pH 7.4, 100 mM KCl, 2 mM MgCl$_2$, 2 mM PMSF, 1% Triton X-100, and 0.6% protease inhibitor cocktail [Sigma]). Lysates were incubated on a rotating wheel at 4°C for 30 min, followed by centrifugation at 17,000× $g$ for 10 min at 4°C. The protein concentration of the supernatant was determined using a Bradford or BCA assay. Biotinylated molecules were isolated by a 1.5 hr incubation of protein samples (between 500 and 1000 µg) with streptavidin beads on a rotating wheel at 4°C. After centrifugation at 3500× $g$, the supernatant was transferred to a fresh tube. Beads were extensively washed, and bound protein was eluted with Laemmli sample buffer with fresh 5% β-mercaptoethanol by heating to 65°C for 15 min. Equal protein amounts of lysates were separated by sodium dodecyl sulfate–polyacrylamide gel electrophoresis (SDS–PAGE) and analyzed by immunoblotting. Bound primary antibodies were detected by incubation with IRDye 680/800CW-conjugated secondary antibodies or, alternatively, HRP-conjugated secondary antibodies and ECL substrate (Pierce 32106). Immunoblots were imaged by LI-COR-Odyssey FC detection with Image Studio Lite Version 4.0. N-cadherin and GAPDH were used as markers for the membrane and cytosol fraction, respectively. All experiments were performed at least four times.

## Analysis of primary neuronal culture extracts

Primary neurons expressing lentivirally delivered either CHC-targeting shRNA or its inactive scramble version were harvested at DIV14, lysed using RIPA buffer (150 mM NaCl, 1% NP-40 al 1%, 0.5% sodium deoxycholate, 0.1% SDS, 50 mM Tris–HCl, protease inhibitor cocktail [Sigma] and phosphatase inhibitor cocktail [Sigma]). Alternatively, primary neurons derived from WT or AP-2µ KO neurons (DIV14) were lysed in HEPES lysis buffer (20 mM HEPES, pH 7.4, 100 mM KCl, 2 mM MgCl$_2$, 2 mM PMSF, 1% Triton X-100, 0.6% protease inhibitor cocktail [Sigma]). Lysates were incubated on a rotating wheel at 4°C for 30 min, followed by centrifugation at 17,000× $g$ for 10 min at 4°C. The protein concentration of the supernatant was determined using a Bradford or BCA assay. Protein samples were denatured and separated in 10% SDS/PAGE followed by western blotting using standard procedures, followed by detection with secondary antibodies coupled to horseradish peroxidase and ECL substrate or IRDye 680/800CW-conjugated secondary antibodies. Immunoblots were imaged and quantified by LI-COR-Odyssey FC detection with Image Studio Lite Version 4.0.

## Statistics

Values are depicted as the mean ± standard error of the mean (SEM) as indicated in the figure legends. For comparisons between two experimental groups, statistical significance was analyzed by two-sample, two-sided unpaired Student's *t*-tests. For comparisons between more than two experimental groups, statistical significance was analyzed by one-way analysis of variance (ANOVA) with a post hoc test such as the Tukey post hoc test (see figure legends). One-sample, two-sided *t*-tests were used for comparisons with control group values that had been set to one for normalization purposes and therefore did not fulfil the requirement of two-sample *t*-tests or one-way ANOVA concerning the homogeneity of variances. GraphPad Prism v.8 software was used for statistical analysis. The level of significance is indicated in the figures by asterisks (*$p \leq 0.05$, **$p \leq 0.01$, ***$p \leq 0.001$, ****$p \leq 0.0001$) and provided in the figure legends as exact p values as obtained by the indicated statistic test. No statistical method was used to predetermine sample sizes as sample sizes were not chosen based on a prespecified effect size. Instead, multiple independent experiments were carried out using several

sample replicates as detailed in the figure legends. Whenever possible, data were evaluated in a blinded manner.

## Contact for reagent and resource sharing

Further information and requests for resources and reagents should be directed to and will be fulfilled by the corresponding contacts V.H. (Haucke@fmp-berlin.de) and S.T. (stakamor@mail.doshisha.ac.jp).

## Acknowledgements

This work was supported by grants from JSPS KAKENHI (19H03330), the JSPS core-to-core program A Advanced Research Networks (grant number: JPJSCCA20170008), a grant from the Takeda Science Foundation to ST, and grants from the Deutsche Forschungsgemeinschaft (SFB958/TP A01; HA2686/13-1/Reinhart-Koselleck Program) to VH. KT is a recipient of The Watanabe Foundation Scholarship grant (grant number: WS2021-013). We are indebted to Sabine Hahn, Delia Löwe, and Silke Zillmann for expert technical assistance with genotyping and preparation of neuronal cultures. We thank Dr. Barth Rossum for his contribution to the illustrations.

## Additional information

### Funding

| Funder | Grant reference number | Author |
| --- | --- | --- |
| Japan Society for the Promotion of Science | KAKENHI (19H03330) | Shigeo Takamori |
| Japan Society for the Promotion of Science | Core-to-Core Program A. Advanced Research Networks (JPJSCCA20170008) | Shigeo Takamori |
| Takeda Science Foundation | | Shigeo Takamori |
| Deutsche Forschungsgemeinschaft | SFB 958/A01 | Volker Haucke |
| Deutsche Forschungsgemeinschaft | HA 2686/13-1 | Volker Haucke |
| The Watanabe Foundation Scholarship grant | WS2021-013 | Koh-ichiro Takenaka |

The funders had no role in study design, data collection, and interpretation, or the decision to submit the work for publication.

### Author contributions

Tania López-Hernández, Conceptualization, Conducted all experiments involving neurons from AP-2μ KO mice and WT littermates, including SEP-based and cypHer5E-based imaging experiments, transferrin uptakes and the detection of surface SV proteins either by biotinylation assays or immunostainings, Data curation, Formal analysis, Investigation, Methodology, Validation, Writing – original draft, Writing – review and editing; Koh-ichiro Takenaka, Conceptualization, Conducted SEP-imaging experiments and analyzed the association of VGLUT1 with adaptor complexes., Data curation, Formal analysis, Investigation, Validation, Writing – original draft, Writing – review and editing; Yasunori Mori, Methodology, Supervision; Pornparn Kongpracha, Shushi Nagamori, Methodology; Volker Haucke, Shigeo Takamori, Conceptualization, Data curation, Funding acquisition, Resources, Supervision, Writing – original draft, Writing – review and editing

### Author ORCIDs

Tania López-Hernández [iD] http://orcid.org/0000-0002-0814-2850
Koh-ichiro Takenaka [iD] http://orcid.org/0000-0001-5600-6565
Yasunori Mori [iD] http://orcid.org/0000-0003-0296-3655

Shushi Nagamori (iD) http://orcid.org/0000-0003-0203-2754
Volker Haucke (iD) http://orcid.org/0000-0003-3119-6993
Shigeo Takamori (iD) http://orcid.org/0000-0003-0215-6124

### Ethics

All animal experiments in the V.H. laboratory and the S.T. laboratory were reviewed and approved by the 'Landesamt f ür Gesundheit und Soziales' (LAGeSo, Berlin) and by the Institutional Animal Care and Use Committee of Doshisha University, respectively, and conducted accordingly to the committee's guidelines.

### Decision letter and Author response

Decision letter https://doi.org/10.7554/eLife.71198.sa1
Author response https://doi.org/10.7554/eLife.71198.sa2

## Additional files

### Supplementary files

• Transparent reporting form

### Data availability

All data generated or analyzed during this study are included in the manuscript and supporting files. Raw images and values are available in Source data files.

The following dataset was generated:

| Author(s) | Year | Dataset title | Dataset URL | Database and Identifier |
|---|---|---|---|---|
| Takamori S | 2021 | Data from: Clathrin-independent endocytic retrieval of SV proteins mediated by the clathrin adaptor AP-2 at mammalian central synapses | http://dx.doi.org/10.5061/dryad.gxd2547mc | Dryad Digital Repository, 10.5061/dryad.gxd2547mc |

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
