## [Editor Report]

Neuronal function requires the recycling of synaptic vesicle proteins from the presynaptic plasma membrane to reform synaptic vesicles. This study highlights a clathrin-independent role of the clathrin adaptor, AP-2 in the endocytic retrieval of select synaptic vesicle cargos at mammalian central synapses. The work will be of interest to cell biologists and neurobiologists interested in understanding the molecular basis of neurotransmission.

---

## [Decision Letter]

**Decision letter after peer review:**

Thank you for submitting your article "Clathrin-independent endocytic retrieval of SV proteins mediated by the clathrin adaptor AP-2 at mammalian central synapses" for consideration by *eLife*. Your article has been reviewed by 3 peer reviewers, and the evaluation has been overseen by a Reviewing Editor and Suzanne Pfeffer as the Senior Editor. The reviewers have opted to remain anonymous.

Essential revisions:

1. All of the reviewers were concerned about the extent of clathrin knockdown, so this will be important to address. The reviewers suggest that you carry out acid quenching for the clathrin RNAi and AP2 KO for an AP2-independent and AP2-dependent protein to show that loss of AP-2 affects endocytosis, not subsequent acidification, which it might also do, although it is hard to understand why this would be specific for certain SV proteins. Quenching surface pHluorin in the absence of clathrin as well as AP-2 should also reveal any effects of both on acidification, which presumably follows SV formation--this would provide an additional positive control for the clathrin knockdown. These do not need to be done for all of these proteins.

2. Please move the transferrin experiment up to Figure 2 as part of a main figure.

*Reviewer #1 (Recommendations for the authors):*

Maybe the authors could speculate more on how AP-2 exerts its function. I am particularly wondering how AP-2 can accumulate the synaptic vesicle proteins in the peri-active zone considering the fact that AP-2 itself undergoes endocytosis. Or in other words, who anchors AP-2? If the authors have any speculative thoughts on this question, it might be worth adding in the discussion.

Line 24: "unified model"

This statement could be toned down a bit. Alternatively, it should be made more explicit in the discussion, in which respect the data provide a unified model.

Line 62: "First"

Maybe mention explicitly which enumeration is to be expected.

Line 542: "The fluorescence intensities following the time course of photobleaching were calculated experimentally for every time point of the recording and summed up to the mean fluorescence intensity at time t."

This sentence is difficult to understand. What is the meaning of "calculated experimentally" or "summed up to the mean"?

Line 669: "of normally distributed data"

The statement is not clear. Were all data normally distributed? Which test of normality was used?

Line 959: "(F)"

Change to (H)

*Reviewer #2 (Recommendations for the authors):*

A role for AP-2 independent of clathrin would be very interesting but the argument relies entirely on the differential effects of AP-2 and clathrin deficiency, and the inactivation of AP-2 (by lentiviral shRNA and cre-dependent knockout) appears more efficient than knockdown of clathrin. In fact, it is a little unclear how strong the knockdown of clathrin is-apparently to 25% by fluorescence but lower than that by western. The amplification involved in immunofluorescence may account for this discrepancy, and the authors show that loss of clathrin does affect endocytosis at room temperature and the transferrin receptor, providing controls for the knockdown. They also examine a range of stimulus conditions and endogenous as well as transfected reporters, increasing confidence in the results. But it would help to clarify the extent of clathrin knockdown or find a way to eliminate it altogether. This is all the more important because there are several implications that are difficult to understand, and the work does not address.

First, AP-2 and clathrin form SVs from an endosomal intermediate, presumably reflecting interactions with many SV proteins. So how does AP-2 promote the endocytosis of only the neurotransmitter transporters?

Second, how does AP-2 promote clathrin-independent endocytosis? Some of the known AP-2 interacting proteins might suggest a mechanism, but positive evidence of a role for machinery operating independent of clathrin would be much more convincing than the absence of a role for clathrin. In addition, some of the data presented here are not new, such as the role of AP-binding sites in endocytosis of the neurotransmitter transporters, and do not add to the accepted view that AP-2 is required for endocytosis, so new information about mechanism would help to understand its role and perhaps selectivity for the transporters, greatly enhancing the impact of the work.

*Reviewer #3 (Recommendations for the authors):*

The authors should address the following concerns:

1. Clathrin. At the outset of this manuscript, appear data indicating that endocytosis of synaptic vesicle proteins is not dependent on clathrin. Even slow processes (operating after 30 seconds) are not clathrin-dependent. This conclusion is followed by data indicating that clathrin adaptors are required. These considerations probably led the authors to include the model in Figure 1, to move the argument away from clathrin to the role of AP-2 as a cargo adaptor. Demonstrating that clathrin is not required is essential for the conclusion that AP-2 is acting independently of clathrin.

A. RNA interference failure at 37degC. The concern is that RNAi for clathrin heavy chain is not effective at physiological temperatures. The control is to demonstrate that clathrin RNAi is effective at blocking transferrin uptake at physiological temperature. That experiment appears in the Supplement to Figure 5. It would be more forceful as a Supplement to Figure 2.

B. RNA interference at 22degC. Instead, the control the authors include in the Supplement demonstrates that clathrin is required for endocytosis of synaptophysin and VGLUT1 at room temperature. They later demonstrate that these two synaptic vesicle proteins require AP-2 for endocytosis, further entangling clathrin and endocytosis. Again, one then wonders if the RNAi at 37{degree sign}C is effective. If the authors have data demonstrating that clathrin RNAi slows endocytosis of synaptotagmin or SV2 at 22{degree sign}C, they should include those data. That would demonstrate that no SV proteins require clathrin at 37{degree sign}C but all require clathrin at room temperature. The requirement for AP-2 is what is variable.

2. Endocytosis vs reacidification. The authors interpret an extended pHluorin signal as a defect in endocytosis, but it may be simply a delay in regeneration of a synaptic vesicle from an endosome-like vacuole. Determining endocytic rates by fluorescence requires rapid quenching. Previously the authors used rapid quenching to demonstrate that 50% of synaptophysin was recovered with a time constant of 700 ms. The issue is important in regard to the AP-2 KO experiments. Is the defect caused by failure to bud synaptic vesicles from the endosome, or is it caused by failure of endocytosis. I recognize that the rapid quenching experiments are not easy; however, it would be useful to demonstrate that rates are similar between synaptophysin and one of the AP-2 dependent synaptic vesicle proteins, and that AP-2 strands the transporters on the surface in their pHluorin assay.

3. Endocytosis during stimulation. Even quenching assays underestimate endocytosis speeds because endocytosis is taking place during the stimulation train. The total fluorescence added to the surface can be determined by stimulating after an acute application of bafilomycin. The authors methodology should capture endocytosis of the readily retrievable pool as well. Together these experiments would provide the best determination of rates of synaptic protein endocytosis to date. Moreover, they represent an improvement over electrophysiological measures given the decrease in health of patch clamp at 37degC. This experiment is only suggested to determine the real speed of endocytosis, and are not essential for the claims of the paper.

4. Time constants. The authors previously found that the rate of endocytosis was fit significantly better by a two-time constants. This represents an easier solution to address the two problems described above.

5. Endocytosis in AP-2 KO. The time constant for initial endocytosis seems rapid even in the absence of AP-2. How are transporters being recovered in the absence of AP-2. It is likely that at least some of this is due to the 20% of AP-2 mu remaining after in these assays. Fitting time constants seems like it can provide insight into this issue as well.

6. Interdependence of vesicle recruitment. The authors demonstrate that AP-2 knockdown slows acidification of VGLUT1, VGAT, synaptophysin and synaptobrevin. Pan et al. (2015) demonstrated that VGLUT1 influences the retrievals kinetics of other cargos, namely synaptophysin and synaptobrevin. It is possible that AP2 is directly involved in the retrieval of VGLUT1, but that synaptophysin and synaptobrevin retrieval kinetics is mediated by VGLUT1. In the AP2 binding mutant of VGLUT1, is the retrieval of synaptophysin and synaptobrevin slowed?

---

## [Author Response]

Essential revisions:1. All of the reviewers were concerned about the extent of clathrin knockdown, so this will be important to address. The reviewers suggest that you carry out acid quenching for the clathrin RNAi and AP2 KO for an AP2-independent and AP2-dependent protein to show that loss of AP-2 affects endocytosis, not subsequent acidification, which it might also do, although it is hard to understand why this would be specific for certain SV proteins. Quenching surface pHluorin in the absence of clathrin as well as AP-2 should also reveal any effects of both on acidification, which presumably follows SV formation--this would provide an additional positive control for the clathrin knockdown. These do not need to be done for all of these proteins.

We have tackled the important issue whether the observed delay in the retrieval of pHluorin (SEP)-tagged VGLUT1 in AP-2 KO neurons is caused by reduced endocytosis kinetics rather than delayed vesicle re-acidification in two distinct ways. First, we used an established quench protocol (e.g. Atluri and Ryan, J Neurosci, 2006) in which acidic buffer is applied before and after neuronal stimulation with 200 APs (new Figure 3M-P) to probe the accessibility of exocytosed VGLUT1- or SV2A-SEP to externally added acid. These experiments showed that exocytosed VGLUT1-SEP accumulates on the surface of AP-2 KO neurons (new Figure 3M) as quantitatively evidenced by an increased DF2/ DF1 ratio (new Figure 3N). No difference in the fraction of surface-accumulated SV2A-SEP molecules was observed in AP-2 KO compared to WT neurons (new Figure 3O,P). In a second set of experiments, we have used a previously published assay (Soykan et al., Neuron 2017) that enables measurements of SV endocytosis independent of re-acidification by probing the acquisition of resistance of SEP fluorescence to externally added acid in the presence of the v-ATPase inhibitor folimycin. Using this assay, we report that even 30 s post-stimulation the vast majority of VGLUT1-SEP has not acquired acid resistance, i.e. remains stranded on the neuronal surface (new Figure 3—figure supplement 1C-F).

Collectively, these new data show that the apparent delay in VGLUT1-SEP quenching observed in conventional fluorescence recordings shown in Figure 3E,F is due to impaired VGLUT1 endocytosis, not a defect in post-endocytic re-acidification.

As for clathrin knockdown neurons, we have opted not to conduct additional technically challenging and time-consuming acid quench experiments, since clathrin-depletion by shRNA did not yield any apparent defects in endocytic decay of any of the six SV proteins analyzed in conventional optical recordings. To further address the possible function of clathrin in SV endocytosis, we conducted additional experiments not requested specifically by the referees to validate the effectiveness of clathrin knockdown. In these experiments, we have combined the best possible clathrin knockdown achievable without compromising neuronal viability with small molecule inhibition of clathrin function by Pitstop 2. The rationale is that clathrin molecules remaining after lentiviral knockdown (about 10-20% of control levels) can still suffice to mediate CME of all six SV cargos, and that the remaining clathrin would be functionally silenced further by pharmacological inhibition. We indeed confirm this prediction by demonstrating that CME of transferrin is potently inhibited down to background levels as shown in the new Figure 5J. In spite of the near complete block of CME, we find that endocytosis of endogenous VGAT probed by cypHer-labeled antibodies proceeds with unperturbed kinetics in response to stimulation with either 50 or 200 APs (new Figure 5F-I). These new data together with our observation (shown in Figure 5K,L) that clathrin knockdown effectively reduces the size of the readily releasable and recycling vesicle pools further corroborate our conclusion that the compensatory endocytosis of SV proteins is independent of clathrin at hippocampal synapses.

2. Please move the transferrin experiment up to Figure 2 as part of a main figure.

As suggested, we now have included the transferrin uptake data as new panels A and B in figure 2 of the revised manuscript. As described above, we have conducted new experiments to address the role of clathrin further using combined pharmacological inhibition of clathrin by Pitstop 2 and its effects on transferrin uptake and endocytosis of endogenous VGAT in clathrin knockdown neurons. The new results confirm that clathrin function is effectively blocked by our manipulations, thereby supporting our conclusion that clathrin is dispensable for SV endocytosis at physiological temperature.

Reviewer #1 (Recommendations for the authors):Maybe the authors could speculate more on how AP-2 exerts its function. I am particularly wondering how AP-2 can accumulate the synaptic vesicle proteins in the peri-active zone considering the fact that AP-2 itself undergoes endocytosis. Or in other words, who anchors AP-2? If the authors have any speculative thoughts on this question, it might be worth adding in the discussion.

We agree that the mechanism of AP-2 localization at sites of SV endocytosis remains an interesting question for future studies and note that even in non-neuronal cells, plasma membrane recruitment of AP-2 occurs independent of clathrin (Motley et al., J Cell Biol 2003). In the revised discussion on p.15, we now outline possible models for the mechanism of AP-2 recruitment and function. We also explicitly spell out that our data in conjunction with previous experiments (Kononenko et al., Neuron 2014) suggest that AP-2 acts at two distinct locations, endocytic sites on the presynaptic surface and on internal ELVs together with clathrin to reform functional SVs.

Line 24: "unified model"This statement could be toned down a bit. Alternatively, it should be made more explicit in the discussion, in which respect the data provide a unified model.

We agree and now use the term "revised model" in the abstract and the last sentence of the introduction.

Line 62: "First"Maybe mention explicitly which enumeration is to be expected.

We have amended the text according to the referee's suggestion.

Line 542: "The fluorescence intensities following the time course of photobleaching were calculated experimentally for every time point of the recording and summed up to the mean fluorescence intensity at time t."This sentence is difficult to understand. What is the meaning of "calculated experimentally" or "summed up to the mean"?

We agree and have modified the corresponding description that should be easier to understand now.

Line 669: "of normally distributed data"The statement is not clear. Were all data normally distributed? Which test of normality was used?

We have re-phrased the corresponding statement.

Line 959: "(F)"Change to (H)

The figure and its corresponding legend have been modified and corrected.

Reviewer #2 (Recommendations for the authors):A role for AP-2 independent of clathrin would be very interesting but the argument relies entirely on the differential effects of AP-2 and clathrin deficiency, and the inactivation of AP-2 (by lentiviral shRNA and cre-dependent knockout) appears more efficient than knockdown of clathrin. In fact, it is a little unclear how strong the knockdown of clathrin is-apparently to 25% by fluorescence but lower than that by western. The amplification involved in immunofluorescence may account for this discrepancy, and the authors show that loss of clathrin does affect endocytosis at room temperature and the transferrin receptor, providing controls for the knockdown. They also examine a range of stimulus conditions and endogenous as well as transfected reporters, increasing confidence in the results. But it would help to clarify the extent of clathrin knockdown or find a way to eliminate it altogether. This is all the more important because there are several implications that are difficult to understand, and the work does not address.

We have taken this crucial point very serious and note that the extent of clathrin knockdown goes to the limits of what can be achieved without severely compromising neuronal viability. To address the issue at stake, we have combined the best possible clathrin knockdown achievable (down to about 10-15% of control levels based on quantitative immunoblotting, Figure 2 – Supplement 1) with small molecule inhibition of clathrin function by Pitstop2. The rationale is that clathrin molecules remaining after lentiviral knockdown (about 10-20% of control levels) are functionally silenced by pharmacological inhibition. We indeed confirm this prediction by demonstrating that CME of transferrin is potently inhibited down to background levels as shown in the new Figure 5J. In spite of the complete block of CME, we find that endocytosis of endogenous VGAT probed by cypHer-labeled antibodies proceeds with unperturbed kinetics in response to stimulation with either 50 or 200 APs (new Figure 5F-I). These new data further corroborate our conclusion that the endocytosis of SV proteins is independent of clathrin at hippocampal synapses.

First, AP-2 and clathrin form SVs from an endosomal intermediate, presumably reflecting interactions with many SV proteins. So how does AP-2 promote the endocytosis of only the neurotransmitter transporters?

Perhaps, our original manuscript has not been clear with respect to the mechanism we envision: We propose that AP-2 (and possibly other endocytic adaptors, e.g. AP180, Stonin 2) operates at the presynaptic cell surface it recognizes and recruit SV proteins such as VGLUT1 and VGAT to sites of endocytosis. The actual endocytic reaction at the neuronal surface can proceed in the absence of clathrin. That said, our earlier data demonstrate that in addition to this activity, AP-2 also operates on internal endosome-like vacuoles (i.e. a distinct location) together with clathrin to reform SVs. This model is now described in more detail on p. 16 of our revised manuscript.

Second, how does AP-2 promote clathrin-independent endocytosis? Some of the known AP-2 interacting proteins might suggest a mechanism, but positive evidence of a role for machinery operating independent of clathrin would be much more convincing than the absence of a role for clathrin. In addition, some of the data presented here are not new, such as the role of AP-binding sites in endocytosis of the neurotransmitter transporters, and do not add to the accepted view that AP-2 is required for endocytosis, so new information about mechanism would help to understand its role and perhaps selectivity for the transporters, greatly enhancing the impact of the work.

In the revised manuscript, we now discuss on p.15 possible models for the recruitment of AP-2 to sites of SV endocytosis on the neuronal surface (e.g. via lipids such as PI(4,5)P_2_ and association with SV cargos such as VGLUT1 and VGAT). We further note that even in non-neuronal cells, plasma membrane recruitment of AP-2 occurs independent of clathrin (Motley et al., J Cell Biol 2003). Hence, our proposal is not inconsistent with previous data.

We agree that some of our data presented are in agreement with prior studies. The crucial point, however, is that we put these previous findings into context and now provide a molecular explanation for the observed physical association of AP-2 with VGLUT1 and VGAT. We hope that the referee agrees that this represents an important conceptual advance and helps to resolve previous discrepancies in the field.

Reviewer #3 (Recommendations for the authors):The authors should address the following concerns:1. Clathrin. At the outset of this manuscript, appear data indicating that endocytosis of synaptic vesicle proteins is not dependent on clathrin. Even slow processes (operating after 30 seconds) are not clathrin-dependent. This conclusion is followed by data indicating that clathrin adaptors are required. These considerations probably led the authors to include the model in Figure 1, to move the argument away from clathrin to the role of AP-2 as a cargo adaptor. Demonstrating that clathrin is not required is essential for the conclusion that AP-2 is acting independently of clathrin.

As suggested, we now have included the transferrin uptake data as new panels A and B in Figure 2 of the revised manuscript.

To further address the possible function of clathrin in endocytosis of endogenous SV proteins such as VGAT, we have conducted additional experiments not requested specifically by the referees. In these experiments we have combined the best possible clathrin knockdown achievable without compromising neuronal viability with small molecule inhibition of clathrin function by Pitstop2. The rationale is that clathrin molecules remaining after lentiviral knockdown (about 10-20% of control levels) are functionally silenced by pharmacological inhibition. We indeed confirm this prediction by demonstrating that CME of transferrin is potently inhibited down to background levels as shown in the new Figure 5J. In spite of the complete block of CME, we find that endocytosis of endogenous VGAT probed by cypHer-labeled antibodies proceeds with unperturbed kinetics in response to stimulation with either 50 or 200 APs (new Figure 5F-I). These new data further corroborate our conclusion that the endocytosis of SV proteins is independent of clathrin at hippocampal synapses.

A. RNA interference failure at 37degC. The concern is that RNAi for clathrin heavy chain is not effective at physiological temperatures. The control is to demonstrate that clathrin RNAi is effective at blocking transferrin uptake at physiological temperature. That experiment appears in the Supplement to Figure 5. It would be more forceful as a Supplement to Figure 2.B. RNA interference at 22degC. Instead, the control the authors include in the Supplement demonstrates that clathrin is required for endocytosis of synaptophysin and VGLUT1 at room temperature. They later demonstrate that these two synaptic vesicle proteins require AP-2 for endocytosis, further entangling clathrin and endocytosis. Again, one then wonders if the RNAi at 37{degree sign}C is effective. If the authors have data demonstrating that clathrin RNAi slows endocytosis of synaptotagmin or SV2 at 22{degree sign}C, they should include those data. That would demonstrate that no SV proteins require clathrin at 37{degree sign}C but all require clathrin at room temperature. The requirement for AP-2 is what is variable.

There appears to be a misunderstanding as to how RNAi is achieved in primary hippocampal neurons. In brief, neurons cultured routinely at 37°C at all times are transduced with lentivirus encoding shRNA targeting clathrin heavy chain at DIV2 and analyzed at DIV14. The same neurons expressing a pHluorin tagged SV protein are then optically recorded at either 22°C or 37°C. At this time point, the lentiviral knockdown (which takes about 10 days to take effect!) has already been completed. The concern that RNAi for clathrin is not effective at physiological temperature is thus unwarranted.

Nevertheless, it should be noted that we indeed show that clathrin knockdown affects CME of transferrin and recycling and readily-releasable vesicle pool sizes, both of which indicate loss of clathrin function at 37°C (new Figure 5J-L, Figure 2—figure supplement 1E).

2. Endocytosis vs reacidification. The authors interpret an extended pHluorin signal as a defect in endocytosis, but it may be simply a delay in regeneration of a synaptic vesicle from an endosome-like vacuole. Determining endocytic rates by fluorescence requires rapid quenching. Previously the authors used rapid quenching to demonstrate that 50% of synaptophysin was recovered with a time constant of 700 ms. The issue is important in regard to the AP-2 KO experiments. Is the defect caused by failure to bud synaptic vesicles from the endosome, or is it caused by failure of endocytosis. I recognize that the rapid quenching experiments are not easy; however, it would be useful to demonstrate that rates are similar between synaptophysin and one of the AP-2 dependent synaptic vesicle proteins, and that AP-2 strands the transporters on the surface in their pHluorin assay.

We have tackled the important issue whether the observed delay in the retrieval of pHluorin (SEP)-tagged VGLUT1 in AP-2 KO neurons is caused by reduced endocytosis kinetics rather than delayed vesicle re-acidification in two distinct ways. First, we used an established quench protocol (e.g. Atluri and Ryan, J Neurosci, 2006) in which acidic buffer is applied before and after neuronal stimulation with 200 APs (new Figure 3M-P) to probe the accessibility of exocytosed VGLUT1- or SV2A-SEP to externally added acid. These experiments showed that exocytosed VGLUT1-SEP accumulates on the surface of AP-2 KO neurons (new Figure 3M) as quantitatively evidenced by an increased DF2/ DF1 ratio (new Figure 3N). No difference in the fraction of surface-accumulated SV2A-SEP molecules was observed in AP-2 KO compared to WT neurons (new Figure 3O,P). In a second set of experiments, we have used a previously published assay (Soykan et al., Neuron 2017) that enables measurements of SV endocytosis independent of re-acidification by probing the acquisition of resistance of SEP fluorescence to externally added acid in the presence of the v-ATPase inhibitor folimycin. Using this assay, we report that even 30 s post-stimulation the vast majority of VGLUT1-SEP has not acquired acid resistance, i.e. remains stranded on the neuronal surface (new Figure 3- Supplement 1C-F).

Collectively, these new data show that the apparent delay in VGLUT1-SEP quenching observed in conventional fluorescence recordings shown in Figure 3E,F is due to impaired VGLUT1 endocytosis, not a defect in post-endocytic re-acidification.

3. Endocytosis during stimulation. Even quenching assays underestimate endocytosis speeds because endocytosis is taking place during the stimulation train. The total fluorescence added to the surface can be determined by stimulating after an acute application of bafilomycin. The authors methodology should capture endocytosis of the readily retrievable pool as well. Together these experiments would provide the best determination of rates of synaptic protein endocytosis to date. Moreover, they represent an improvement over electrophysiological measures given the decrease in health of patch clamp at 37degC. This experiment is only suggested to determine the real speed of endocytosis, and are not essential for the claims of the paper.

We agree that determining the real speed of SV endocytosis using fast perfusion acid-quenching experiment is interesting and was conducted in our earlier 2017 study (Soykan et al., 2017). We hope that the referee agrees that repeating these technically challenging and laborious experiments here goes beyond the scope of the present manuscript. We also note that the focus of our study was to find out whether AP-2 function is limited to SV reformation from internal structures or, whether AP-2, in addition, can sort select SV cargo at the neuronal surface. We believe that our experiments provide strong evidence that this is the case.

4. Time constants. The authors previously found that the rate of endocytosis was fit significantly better by a two-time constants. This represents an easier solution to address the two problems described above.

We indeed have previously used double exponentials to fit endocytosis rates determined by rapid acid quenching experiments of neurons stimulated with either 2 or 10 APs (Soykan et al., 2017). Given the caveat that ultrafast endocytosis cannot be detected by conventional pHluorin imaging technology (and likely would be a minor component, at least for 200 AP trains), we prefer to fit the observed fluorescence decays shown in Figures 2-4 and the rise in Figure 5 by single exponential functions. Previous attempts to fit such curves by double exponential functions in our opinion have not resulted in improved fits.

5. Endocytosis in AP-2 KO. The time constant for initial endocytosis seems rapid even in the absence of AP-2. How are transporters being recovered in the absence of AP-2. It is likely that at least some of this is due to the 20% of AP-2 mu remaining after in these assays. Fitting time constants seems like it can provide insight into this issue as well.

We indeed go to the highest possible efficacy of AP-2m depletion possible without severely compromising neuronal viability. The referee is right that we cannot exclude functional effects of the low number of remaining AP-2 molecules on VGLUT1 or VGAT endocytosis. That said, we kindly refer the referee to our analysis of the apparent rate of endocytosis of endogenous VGAT in Figure 5B-E. From this analysis, it appears that the initial rate of VGAT internalization is slowed quite substantially in the near absence of AP-2.

We also now discuss on p. 16 of the revised manuscript the possibility that confinement of exocytosed SV proteins including neurotransmitter transporters may account for the endocytic retrieval of SV proteins in the absence of their specific sorting adaptors (e.g. VGLUT1 or VGAT endocytosis in AP-2 KO neurons).

6. Interdependence of vesicle recruitment. The authors demonstrate that AP-2 knockdown slows acidification of VGLUT1, VGAT, synaptophysin and synaptobrevin. Pan et al. (2015) demonstrated that VGLUT1 influences the retrievals kinetics of other cargos, namely synaptophysin and synaptobrevin. It is possible that AP2 is directly involved in the retrieval of VGLUT1, but that synaptophysin and synaptobrevin retrieval kinetics is mediated by VGLUT1. In the AP2 binding mutant of VGLUT1, is the retrieval of synaptophysin and synaptobrevin slowed?

We agree and have discussed this possibility on p.15 of the revised manuscript.